# Selective single molecule sequencing and assembly of a human Y chromosome of African origin

Lukas F.K. Kuderna [1], Esther Lizano [1], Eva Julià[2,3], Jessica Gomez-Garrido[4], Aitor Serres-Armero[1], Martin Kuhlwilm [1], Regina Antoni Alandes[4], Marina Alvarez-Estape[1], David Juan[1], Heath Simon [4,5], Tyler Alioto [4,5], Marta Gut[4,5], Ivo Gut[4,5], Mikkel Heide Schierup[6,7], Oscar Fornas [3,5] & Tomas Marques-Bonet [1,4,5,8,9]

Mammalian Y chromosomes are often neglected from genomic analysis. Due to their inherent assembly difficulties, high repeat content, and large ampliconic regions, only a handful of species have their Y chromosome properly characterized. To date, just a single human reference quality Y chromosome, of European ancestry, is available due to a lack of accessible methodology. To facilitate the assembly of such complicated genomic territory, we developed a novel strategy to sequence native, unamplified flow sorted DNA on a MinION nanopore sequencing device. Our approach yields a highly continuous assembly of the first human Y chromosome of African origin. It constitutes a significant improvement over comparable previous methods, increasing continuity by more than 800%. Sequencing native DNA also allows to take advantage of the nanopore signal data to detect epigenetic modifications in situ. This approach is in theory generalizable to any species simplifying the assembly of extremely large and repetitive genomes.

[1] Institut de Biologia Evolutiva, (CSIC-Universitat Pompeu Fabra), PRBB, Doctor Aiguader 88, Barcelona, Catalonia 08003, Spain. [2] Institut Hospital del Mar d'Investigacions Mèdiques (IMIM), Carrer del Doctor Aiguader 88, PRBB Building, Barcelona 08003, Spain. [3] Centre for Genomic Regulation (CRG), The Barcelona Institute for Science and Technology, Carrer del Doctor Aiguader 88, Barcelona 08003, Spain. [4] CNAG-CRG, Centre for Genomic Regulation (CRG), The Barcelona Institute of Science and Technology, Baldiri Reixac 4, Barcelona 08028, Spain. [5] Universitat Pompeu Fabra (UPF), Doctor Aiguader 88, Barcelona 08003, Spain. [6] Bioinformatics Research Center, Aarhus University, C.F. Moellers Alle 8, DK-8000 Aarhus C, Denmark. [7] Department of Bioscience, Aarhus University, Ny Munkegade 116, DK-8000 Aarhus C, Denmark. [8] Institució Catalana de Recerca i Estudis Avançats (ICREA), Passeig Lluís Companys 23, Barcelona, Catalonia 08010, Spain. [9] Institut Català de Paleontologia Miquel Crusafont, Universitat Autònoma de Barcelona, Edifici ICTA-ICP, c/ Columnes s/n, Cerdanyola del Vallès, Barcelona 08193, Spain. These authors contributed equally: Lukas F.K. Kuderna, Esther Lizano, Oscar Fornas, Tomas Marques-Bonet. Correspondence and requests for materials should be addressed to L.F.K.K. (email: lukas.kuderna@upf.edu) or to E.L. (email: esther.lizano@upf.edu) or to T.M-B. (email: tomas.marques@upf.edu)

Recombinational arrest in the common ancestor of the X and Y chromosomes led to the degeneration and accumulation of large amounts of repetitive DNA on the Y chromosome along its evolutionary trajectory[1]. Furthermore, many sequencing efforts have traditionally chosen female samples, as the hemizygous nature of the sex chromosomes leads to half the effective sequencing coverage on both of them in a male, resulting in inferior genome assemblies[2,3]. Together, these causes have led to an underrepresentation of Y chromosomes in genomic studies and proper characterization of the Y chromosome in only a handful of mammalian species through a time- and labor-intensive clone by clone approach[4–7]. One strategy to reduce the complexity of the assembly problem for the Y chromosome is to isolate it by flow cytometry, thus dramatically reducing the potential amount of overlaps of repetitive regions in the context of the whole genome[3]. Notwithstanding previous efforts, which sought to do this have faced some drawbacks, as the material has been heavily amplified post sorting to increase yield[8]. Whole-genome amplification (WGA) introduces biases that are detrimental to genome assembly, such as unequal sequence coverage and chimera formation, as well as limited fragment length[9]. Moreover, these methods lead to the loss of epigenetic modifications that can now be directly determined from the signal data from nanopore sequencers[10]. Additionally, previous efforts

to assemble the Y chromosome purely from flow-sorted material did so using the gorilla[8], a species with a previously uncharacterized Y chromosome, meaning that potential biases in the assembly cannot be detected without a gold standard reference to compare with, such as human. Integrating single-molecule sequencing has been shown to produce far superior whole-genome shotgun (WGS) assemblies than sequencing by synthesis platforms[11–14]. Furthermore, the MinION sequencing platform from Oxford Nanopore Technologies has recently been used to create the most contiguous human WGS assembly to date[15] and to resolve the structure of the human Y-chromosome centromere[16]. To take advantage of these benefits, we developed a protocol to sequence native, unamplified flow-sorted DNA on the MinION sequencing device.

## Results

**Flow sorting and sequencing.** We sorted approximately 9,000,000 individual Y chromosomes from a lymphoblastoid cell line (HG02982) from the 1000 Genomes Project, whose haplogroup (A0) represents one of the deepest known splits in humans[17] (see Fig. 1a). Given the large volume in which the chromosomes were sorted, and potential issues with residual dyes that are necessary for the sorting process, we devised a purification protocol to bring the

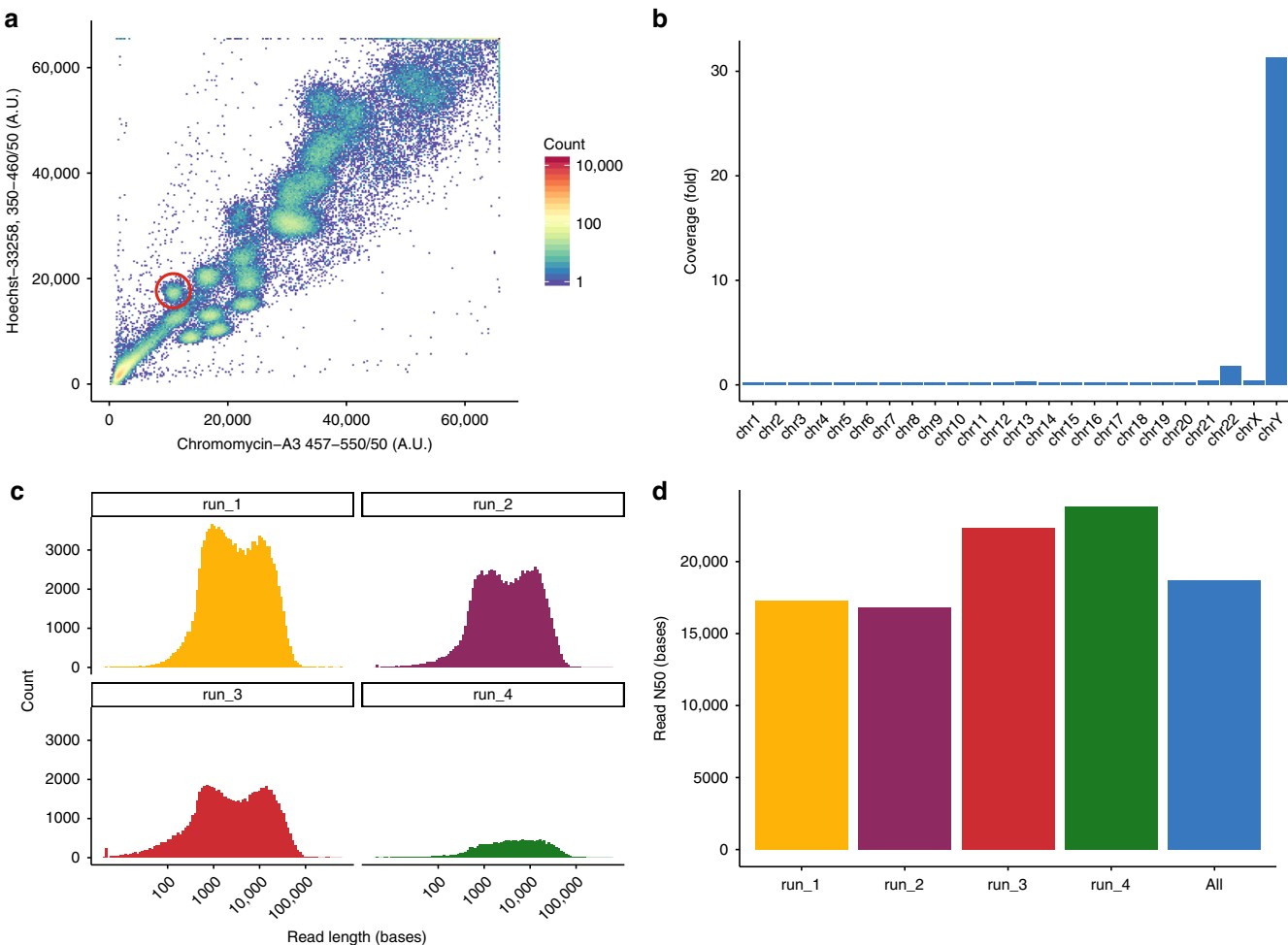

**Fig. 1** Flow-sorting and sequencing specificity. **a** Flow-karyogram of a human genome. The different clusters correspond to different chromosomes. The red circle delimits the cluster corresponding to the Y chromosome used for this project. **b** Enrichment specificity of the sequencing data. Sequences on the Y chromosome are ~110-fold enriched compared with WGS sequencing. Chromosome 22 partially co-sorts with Y. All other chromosomes are depleted. **c** Read length (log10 scale) distribution of the four runs. **d** N50 values for all four runs and the combined dataset. Colors in panels **c** and **d** correspond to the different runs. Source data are provided as a Source Data file

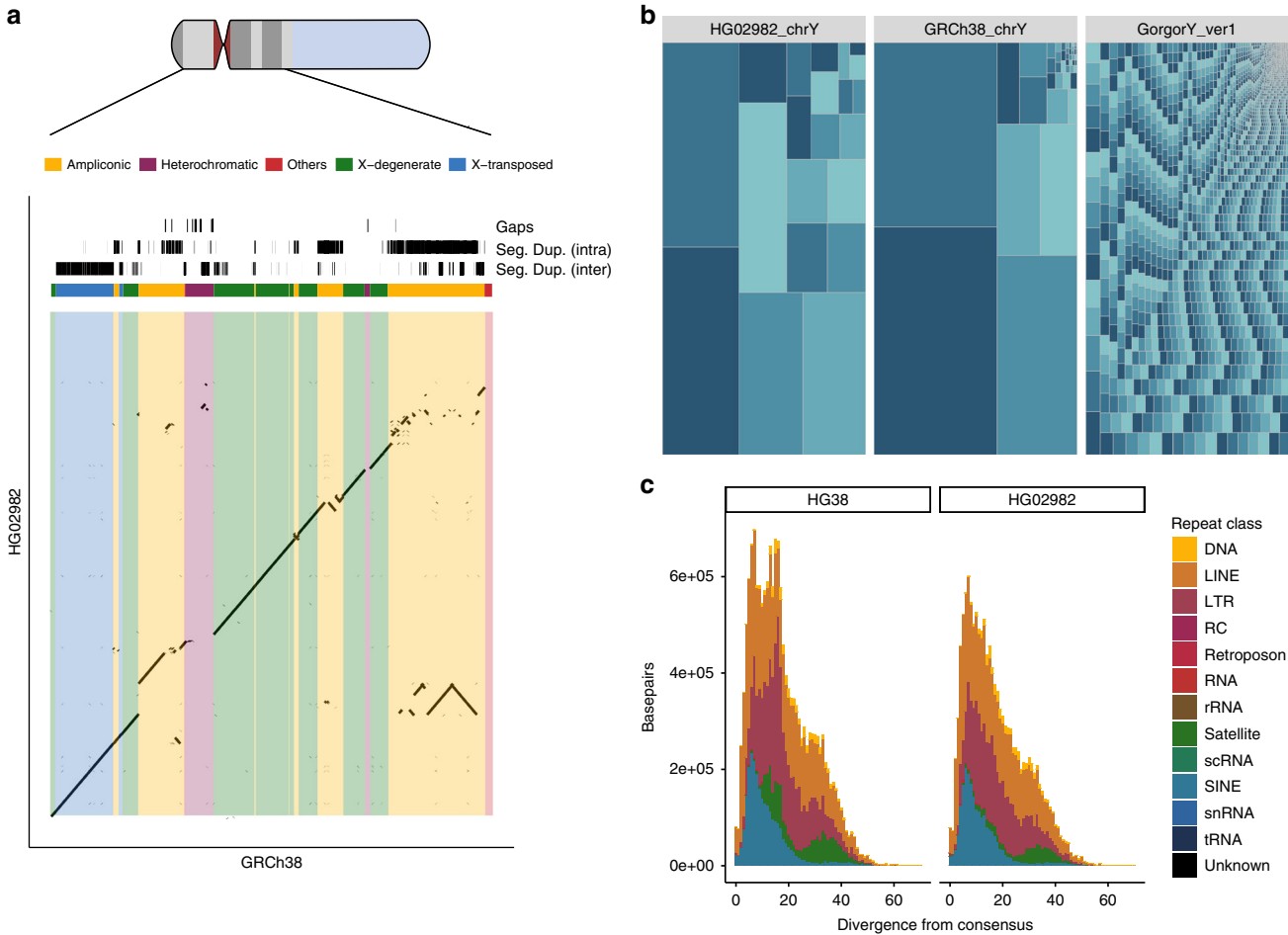

**Fig. 2** Chromosome-Y assembly overview and comparisons. **a** Dot-plot comparing the resolved MSY of GRCh38 with HG02982. The reconstruction is highly continuous along most sequence classes, with ampliconic regions showing a higher degree of fragmentation. Seg. Dup. (intra) refers to intra-chromosomal segmental duplications, Seg. Dup. (inter) refers to inter-chromosomal segmental duplications. Altogether, ~ 50% of the of the Y chromosomes resolved sequence space in GRCh38—> 13 Mb—are annotated as segmental duplications. **b** Treemap comparing the contiguity of HG02982 chrY with GRCh38 chrY and the gorilla Y chromosome by Tomaszkiewicz et al. The size of each rectangle corresponds to the size of a contig within each of the assemblies. Neighboring rectangles are colored differently as a visual aid. **c** Repeat landscape of common, interspersed repeats annotated equally in GRCh38 and HG02982. Common repeats—including very recent ones—are well resolved in HG02982. The exception are satellite sequences, and a population of somewhat divergent (~ 20%) LTR elements, which are absent in HG02982 (see supplementary Figures 7-9). Source data are provided as a Source Data file

DNA into conditions suitable for sequencing. We ran four Oxford Nanopore MinION flowcells to generate 305,528 reads summing to over 2.3 Gb of data. The yields per flowcell varied considerably from 897.6 Mb to 163.8 Mb (see Fig. 1c and Supplementary Table 2). Sequencing yields were on the lower end of the reported spectrum (75 Mb–5.5 Gb per flowcell in[15]), but read N50 surpassed most of them, ranging from 16.8 to 23.8 kb[15] (see Fig. 1d, supplementary Figures 1–2). Additionally, for the same flow-sorted material we ran an Illumina MiSeq lane for $2 \times 300$ cycles but including four rounds of PCR amplification. To check the enrichment specificity, we aligned the reads to the human reference genome (GRCh38) and calculated the normalized coverage on each chromosome. Taking into account the size of the Y chromosome and its haploid nature, we find it to be over 110-fold enriched compared with a random sampling from the human genome (see Fig. 1b, Supplementary Figures 3–5, Supplementary Note 2 and Supplementary Data 1–2).

**Y-chromosome assembly and comparison with GRCh38.** We used the Nanopore data to construct a de novo assembly using

Canu[18]. We performed a self-correction by aligning the reads used for assembly and called consensus using Nanopolish[10], correcting a total of 127,809 positions. Finally, the Illumina library served to polish residual errors within the assembly using pilon[19]. By this means, we corrected a further 101,723 single-nucleotide positions and introduced 105,640 small insertions and 6983 small deletions. We also explored further polishing options and found that running one additional round of error correction with racon[20] potentially resolves several remaining errors, despite also introducing additional discordances (see Supplementary Table 4, Supplementary Notes 1,3 and Supplementary Figures 10–13). The final assembly is comprised of 35 contigs, with an N50 of 1.46 Mb amounting to 21.5 Mb of total sequence, in contrast to a contig N50 of 6.91 Mb of the GRCh38 Y-chromosome assembly. Compared with the gorilla Y-chromosome assembly with a contig N50 of 17.95 kb[8], our assembly is two orders of magnitude more contiguous (see Fig. 2b).

The Y chromosome is comprised of a set of discrete sequence classes[4]. To check the completeness of our assembly, we assessed how well each of them is represented. After retaining only single best placements, we were able to align 21.1 Mb, or 98.4% of its

**Table 1 Assembly statistics overview**

| Seq. class | Aln. HG02982 (b) | HG02982 ID SNP (%) | HG02982 ID SNP + InDel(%) | Rec. HG02982 (%) | Aln. NA24385 (b) | Rec. in NA24385 (%) | Len. w/o gaps (b) |
|---|---|---|---|---|---|---|---|
| Ampliconic | 6,146,087 | 99.91 | 99.67 | 62.67 | 5,242,461 | 53.46 | 9,807,089 |
| Heterochromatic | 543,005 | 99.66 | 99.31 | 32.77 | 171,045 | 10.32 | 1,656,797 |
| Others | 295,160 | 99.47 | 99.18 | 385.59 | 63,973 | 83.57 | 76,547 |
| Pseudo-autosomal | 2,219,743 | 99.58 | 99.13 | 78.02 | 117,626 | 4.13 | 2,844,939 |
| X-degenerate | 8,537,493 | 99.95 | 99.81 | 98.94 | 8,238,733 | 95.48 | 8,628,904 |
| X-transposed | 3,374,011 | 99.94 | 99.81 | 99.21 | 1,474,610 | 43.36 | 3,400,750 |

Summary of sequence class coverage of HG02982 versus GRCh38, as well as the contigs from NA23385 identified as derived from the Y chromosome. The proportion of recovered sequences and % identity are calculated over the resolved sequences in GRCh38, excluding gaps. There are currently 30.8 Mb of unresolved sequence (represented by the ambiguous base N) in the reference Y chromosome of GRCh38, the vast majority of which belongs to heterochromatin on the q arm
Aln.: aligned bases to GRCh38, ID.: percent identical bases in GRCh38, Rec.: recovered proportion from GRCh38, Len.: length in GRCh38

total length, with 99.9% of identical bases on average (see Fig. 2a). We recovered the full-length (~ 99% of the annotated length in GRCh38) reconstructions of both the X-transposed and the X-degenerate regions. Although the X-degenerate region can be considered a single-copy region due its distant common ancestry with the X chromosome, the X-transposed region emerged only after the split between humans and chimpanzees[21]. The largest sequence class on the Y chromosome is comprised of ampliconic regions, which amount to around 30% of the euchromatic portion and sum to 9.93 Mb. These regions contain eight massive, segmentally duplicated palindromes, all of which share >99.9% identity between their two copies, with the largest one spanning over 2.90 Mb. We find this region to be the most challenging to reconstruct, with fragmented and collapsed sequences, but are nevertheless able to recover 6.14 Mb, or 62.7% of its length in GRCh38. Surprisingly, we recover only 78% of the pseudo-autosomal regions (PARs). We observed a rather steep drop-off in coverage coinciding with the PAR-1 boundary on GRCh38. As we are sequencing native, unamplified DNA, the genomic coverage is directly proportional to the number of copies of the underlying sequenced region[22]. We compared the mapped coverage of our raw data on GRCh38 and find that PAR-1 exhibits only around 72% of the average coverage of the whole chromosome (19.8-fold versus 27.3-fold). We observe the drop-off in coverage to coincide sharply with the PAR-1 boundary (see Supplementary Figure 6). Finally, of the remaining sequence classes, we are able to recover around 32.8% of the resolved heterochromatic regions, and multiple instances of the remaining unclassified sequences (referred to as other; see Table 1 and Supplementary Data 3).

To contrast our approach to a long-read WGS assembly, we assembled the publicly available PacBio dataset from the Ashkenazim son from the Genome in a Bottle Consortium[23], which has a sequencing depth comparable to ours on the sex chromosomes (~ 30X). We identified 193 contigs mapping to the Y chromosome, with an N50 of 213 kb, covering 15.3 Mb, or around 28% less than by our approach. The WGS fails to assemble roughly 56.6% of the X-transposed region and 47% of the ampliconic regions (see Table 1).

**Comparative gene annotation**. We performed a comparative annotation to check the completeness of our assembly at the gene level. To this end, we projected all Gencode (v. 27, GRCh38) annotations on the Y chromosome onto our assembly and annotated them there. Due to its peculiar evolutionary trajectory, the gene-space on the Y chromosome is degenerated, and any remaining genes can generally be classified into two categories: on one hand there are single-copy genes, which are broadly expressed beyond the testis. On the other, there are multi-copy genes within the ampliconic regions, which are mainly involved in spermatogenesis[24]. We recover the complete gene set of the genes in the

male-specific region of the Y chromosome (MSY) region and are therefore able to annotate all single-copy genes. Furthermore, we are able to retrieve at least one member of all multi-copy gene families. For four out of nine of these gene families, we are additionally able to resolve further copies within our assembly (see Supplementary Data 4–5). We also note that four genes (ASMTL, IL3R, P2RY, SLC25) from a comparatively short syntenic block of around 200 kb are partially missing from our assembly due to the aforementioned technical challenges in the PAR-1 region. Mapping the raw data onto GRCh38 show that this is an artifact, presumably due to insufficient coverage in this region.

**Structural variants**. We produced a stringent call set of structural variants (SVs) derived from alignments to GRCh38 using Assemblytics[25]. We detect 347 SVs at least 50 bp in size (931 variants at least 10 bp in size) of which 82 are at least 500-bp long (see Fig. 2c, Supplementary Figures 14–15, Supplementary Table 3). The cumulative length of these variants sums to 184 kb. We observe a 4.8-fold excess number of deletions versus number of insertions, amounting to a twofold excess of bases in deletions versus bases in insertions. Although a deletion bias for nanopore-based assemblies had previously been reported[15], we find the strength of this bias to be decreasing in our analysis, probably reflecting improvements in base-calling accuracy. To check the presence of large-scale copy number variation in multi-copy genes, we additionally determined the chromosome-wide copy number based on a read depth approach using the Illumina data. We find extensive genic copy number variation, with expansions in five of the nine multi-copy genes, when compared with the reference individual. Among these, we find expansions in RBMY, PRY, BPY2, and DAZ, all members of the AZFc region locus with implications for male fertility. Although these expansions are to some degree represented in our assembly, the precise genomic architecture remains challenging to reconstruct. Due to the high degree of similarity between copies, several of them will be collapsed in the assembly specially in the AZFc region. Finally, to assess concordance with previous studies, we compared our SV calls with those generated by the 1000 Genomes Project, which contains the same cell line used for this study[26]. We manually confirm the presence of all structural three variants called in HG02982 in the 1000 Genomes Project in our data by checking the overlap of calls produced by orthogonal approaches (see Supplementary Figures 16–18).

**CpG methylation status**. Finally, we called the methylation status of 5-methylcytosines (5-mC) at CpG positions from the Nanopore signal data using a recently developed model implemented in Nanopolish[10]. To assess potential biases on the CpG methylation status introduced by our workflow, we also produced whole-genome bisulfite sequencing data (WGBS) for the same cell

line. We calculated the methylation frequency (i.e., the proportion of reads supporting 5-mc at a given CpG) for both datasets. For positions where both datasets have at least 10-fold coverage ($n = 4654$), we observe a good concordance in the methylation frequency with a Pearson's $r$ of 0.816 (see Supplementary Figure 19 and Supplementary Table 5). Remaining differences might be attributable to differences in sensitivity, variation in the methylation state, or alternative modifications such as 5-hydroxymethylation, which cannot be distinguished from 5-mC by WGBS[10]. Additionally, detecting the 5-mC status on the Y chromosome from long reads in our methodology has the advantage of allowing to interrogate regions that are not accessible to WGBS with short reads, namely the PAR, the X-transposed region, and to some degree the Ampliconic regions. We interrogated the methylation state of CpG 200-bp upstream of the transcription start site (TSS) in protein-coding genes falling within the different sequence classes of the Y chromosome. Genes from the PAR, X-degenerate, and X-transposed regions are expressed throughout the body, whereas Ampliconic genes have testis specific expression[24]. In agreement with these patterns, we find the genes within PAR, X-degenerate, and X-transposed regions to show low degrees of CpG methylation at TSS. Within the Ampliconic regions, the distribution of methylation frequencies of CpGs at TSS shows an overall high degree of methylation and is therefore consistent with the expected downregulation of these genes in lymphoblastoid cells (see Supplementary Figures 20–21). Nevertheless, single-copy resolution is not possible due to potential mapping ambiguities.

## Discussion

Here, we report the first successful sequencing and assembly of native, flow-sorted DNA on an Oxford Nanopore sequencing device, without previous amplification. We apply our methodology to assemble the first human Y chromosome of African origin to benchmark our approach. This is arguably the most challenging human chromosome to assemble due to its high repeat and segmental duplication content, and hence a good test-case to explore the possibilities and limitations of this approach. With the exception of bacterial artificial chromosome-based assemblies, we are able to reconstruct the Y chromosome to unprecedented quality in terms of contiguity and sequence class representation. We show that we not only outperform previous efforts that sought to achieve a similar goal of reconstructing Y chromosomes[8], but also accomplish a better reconstruction on all sequence classes than the Y chromosomal sequences derived from a long-read WGS assembly. Additionally, our method is orders of magnitude cheaper than reconstructions from WGS data too, especially considering that twice the desired Y chromosomal target coverage is needed on the autosomes. Given the current developments in sequencing throughput, a single-MinION flow-cell should now be sufficient to assemble a whole human Y chromosome. Furthermore, it is becoming clear that the upper read length boundary is only delimited by the integrity of the DNA, suggesting the possibility that complete Y-chromosome assemblies, including full resolution of amplicons, might be possible in the near future. Notwithstanding, some challenges to obtain ultra-long reads from flow-sorted chromosomes are still to be overcome, as sorting sufficient material for this protocol is a substantial endeavor. It also is worth noting that our efforts to sequence the same input material on Pacific Biosciences Sequel platform have been fruitless, presumably due to interference of residual dyes with the sequencers optical detection system. Despite the technical challenges of flow-sorting single chromosomes, the method described here offers the opportunity to take advantage of the benefits of long-range data together with local

complexity reduction. Given different chromosomes that are sufficiently distinguishable in terms of size and GC content, immediate applications are either very complex chromosomes, such as the human Y, or extremely large genomes with a very high degree of common repeats, which have long challenged traditional WGS approaches, such as wheat, the loblolly pine, or the axolotl[27–30].

## Methods

**Chromosome preparation for flow karyotyping.** Mitotic chromosomes in suspension were prepared as follows (adapted from[31] with some modifications): the lymphoblastoid cell line HG02982 (purchased from Coriell, cat. no. HG02982) were cultured in RPMI-1640 medium supplemented with 2 mM L-glutamine (Invitrogen, ref. 21875-034), 15% fetal bovine serum and antibiotics (penicillin and streptomycin (Invitrogen, ref. 15140-122)) at initial concentration no <150,000 viable cells per ml. Near confluence, cells were subcultured to 50%. After 24 h, the cells were blocked in mitosis by adding Colcemid to the culture (10 μg ml$^{-1}$ demecolcine solution (Gibco, ref. 15210-040)) to a final concentration of 0.1 μg ml$^{-1}$ and incubated for an additional 6–7 h. To swell and stabilize mitotic cells, they were centrifuged 5 min at $300 \times g$ at room temperature. The pellet was slowly resuspended in 10 ml hypotonic solution (Hypotonic solution: 75 mM KCl, 10 mM MgSO$_4$, 0.2 mM spermine, 0.5 mM spermidine. pH 8.0), incubated for 10 min at room temperature. After the incubation in the hypotonic solution, the swollen cells were centrifuged at $300 \times g$ for 5 min. The cell pellet was resuspended in 1.5 ml of ice-cold polyamine isolation buffer (PAB: 15 mM Tris, 2 mM EDTA, 0.5 mM EGTA, 80 mM KCl, 3 mM dithiothreitol, 0.25% Triton X-100, 0.2 mM spermine, 0.5 mM spermidine. pH 8.0) for 20 min to release the chromosomes.

To ensure the integrity of the chromosomes, their morphology was checked before staining them. To this end, the pellet was vigorously vortexed for 30 s to liberate the chromosomes from the mitotic cells. The suspension was filtered through a 35 μm mesh filter and stored at 4 °C until its sorting.

Finally, chromosomes were stained with chromomycin-A3 (Sigma, ref. C2659) and Hoechst 33,258 (Invitrogen, ref. H3569) at a final concentration of 40 μg ml$^{-1}$ and 5 μg ml$^{-1}$, respectively, in presence of divalent cations (10 mM MgSO$_4$ (Sigma, ref. 60142)). Staining was performed for at least 8 h at 4 °C, to allow the dyes to equilibrate. Before the sample analysis on a cell sorter, potassium citrate was added to a final concentration of 10 mM (Sigma, ref. 89306) to enhance peak resolution in the flow karyotype.

**Chromosome sorting.** Flow karyotyping for chromosome sorting was performed on BD Influx cell sorter (Becton Dickinson, San Jose, CA), a jet-in-air cell sorter that was selected for its relatively easy manual daily fine-tuning and high-resolution capabilities. Of the five available lasers, only the blue (488 nm laser at 200 mW), deep-blue (457 nm laser at 300 mW), and ultraviolet (355 nm laser at 100 mW) ones were used for flow karyotyping. The setup and performance were optimized using standard 8-peaks Rainbow beads (Sphero$^{TM}$ Rainbow Calibration Particles 3.0–3.4 μm, BD Biosciences, ref. 559123), 1-peak UV beads for UV laser alignment (Alignflow$^{TM}$ Flow Cytometry Alignment 2.7 μm, Molecular Probes, ref. A16502), and 1-peak 457 nm for deep-blue laser alignment (Fluoresbrite$^{TM}$ Plain YG Microspheres 1.0 μm, Polysciences, Inc. ref. 17154) were, respectively, used for 488-blue, 355-UV, and 457-deep-blue optimal laser alignment and instrument fine tuning to obtain the highest resolution of chromosome detection and sorting.

The threshold for chromosome sorting was set triggering in chromomycin-A3 fluorescence on 457 nm laser as primary excitation line and set at approximately 1800 a.u. Then, chromomycin-A3 fluorescence was used as primary fluorescence reference through a light line of 500 LP filter and collected by a 550/50 nm band-pass filter. Hoechst was excited with the UV laser and its fluorescence was collected through a light line of 400 LP filter and by 460/50 BP. All parameters were collected in lineal mode and analyzed with the BD FACS$^{TM}$ Software (v. 1.0.0.0.650, Becton Dickinson, San Jose, CA).

We chose a 100 μm nozzle because we found it to have the best piezoelectric frequency/electronic-noise ratio. The piezoelectric frequency was adjusted at 38.7 KHz. The sample flow rate for chromosome sorting was adjusted at up to 6000 events s$^{-1}$. The gating strategy for chromosome sorting was simple because only a bi-parametrical dot-plot Hoechst versus chromomycin-A3 fluorescence was used (see Fig. 1a).

**Purification and concentration of flow-sorted Y chromosomes.** For each of the two rounds of purification, the fractions corresponding to approximately 4.5 M Y chromosomes (~ 500 ng of DNA per aliquot) were divided into 1 ml aliquots with an estimated chromosome count of 400,000, corresponding to a DNA concentration of approximately 0.04 ng μl$^{-1}$. The approximate total volume per round of purification was around 22.5 ml. Each tube containing the flow-sorted DNA was treated overnight with 10 μl of proteinase K (20 mg ml$^{-1}$) at 50 °C. After treatment, the buffer was exchanged, and proteinase K, as well as chromomycin-A3 and Hoechst 33,258 removed by dialysis against 1 liter of TE buffer using a Pur-A-Lyzer$^{TM}$ Maxi Dialysis column with a molecular weight cut-off of 50 kDa (Sigma-Aldrich). Dialysis was carried out for 48 h exchanging the buffer every 10–16 h. To

reduce the volume after buffer exchange, DNA was transferred into 1.5 ml tubes and concentrated by evaporation in a miVac DNA concentrator (Barnstead GeneVac, Ipswich, UK) up to a volume of approximately 5–10 µl. A final purification step was performed by pooling the concentrated DNA into two tubes and subjecting it to a solid-phase reversible immobilisation (SPRI) bead purification with a 2X ratio (SPRI beads/sample). DNA was eluted in 9 µl of low TE buffer and pooled into one tube. Concentrations were determined by absorbance at 260 nm with a NanoDrop 2000 (Thermo Scientific) and by fluorometric assay with the Qubit 2.0 using the Qubit dsDNA HS kit (Invitrogen) (see Supplementary Table 1).

**Sequencing the flow-sorted chromosomes.** The purified DNA was prepared for sequencing following the protocol in the Rapid Sequencing kit SQK-RAD002 (ONT, Oxford, UK). Briefly, approximately 200 ng of purified DNA was tagmented for 1 min at 75 °C with the Fragmentation Mix (ONT, Oxford, UK). The Rapid Adapters (ONT, Oxford, UK) were added along with Blunt/TA Ligase Master Mix (NEB, Beverly, MA) and incubated for 30 min at room temperature. The resulting library was combined with Running Buffer with Fuel (ONT, Oxford, UK) and Library Loading Beads (ONT, Oxford, UK) and loaded onto a primed R9.4 Spot-On Flow cell (FLO-MIN106). Sequencing and initial base calling was performed with a MinION Mk1B MinKNOW v1.7.10 software package running for 48 h. Estimates for DNA quantification were based on chromosomal counts with corresponding quantification values from Gribble et al.[31]. The uncertainties in quantification with Qubit 2.0 or NanoDrop are presumed to be due to residual intercalating dyes present within the sample, which interfere with the quantification platforms detection systems, with competition of additional intercalants leading to underestimation on the Qubit 2.0, and the additional presence of aromatic groups leading to overestimation on the NanoDrop.

A total estimated amount of 100 ng of Y chromosome was fragmented on a Covaris ultrasonicator with settings targeting fragments of 450 bp. The library was prepared using NEBNext Ultra II DNA Library Prep Kit (New England BioLabs) following the manufacturer's instructions, including four cycles of PCR amplification. Agilent BioAnalyzer High-Sensitivity DNA Kit was used to determine the size distribution and molarity. The library was sequenced on an Illumina MiSeq using the v3 kit and 600 cycles resulting in 300-bp paired-end reads.

**Assembly, error correction, and polishing.** The initial base-calls (MinKNOW 1.7.10 using Albacore 1.1) from the Nanopore data were assembled with Canu (v 1.6)[18] without previous read separation of reads deriving from different chromosomes and assuming a chromosome size of 52 Mb. The following parameters were used:

```
canu -p HG02982 -d HG02982_canu genomeSize=52
m overlapper=mhap utgReAlign=true -nanopore-
raw raw_data/HG02982/all.joint.fastq
```

The 2.3 Gb of input data resulted in 25X of error corrected reads for assembly, assuming a chromosome size of 52 Mb. The data assembled into 35 contigs, which where self-corrected using the Nanopore input reads. To this end, we re-performed base calling from the fast5 files using Albacore (v 2.1, available from the nanopore user community) to be used for variant calling with Nanopolish (v. 0.8.4, https://github.com/jts/nanopolish, 11 December 2017).

```
read_fast5_basecaller.py -f FLO-MIN106 -k
SQK-RAD002 -i input_folder -s outout_folder -t 8
-o fastq,fast5 -q 10000000 -n 100000 --disable_
pings
```

We indexed the reads to be used with Nanopolish:

```
nanopolish index -f fast5.fofn reads.joint.
fastq
```

The reads were mapped onto the raw assembly using bwa mem (v. 0.7.120)[32] with the additional flag -x ont2d and the mappings merged and sorted with samtools (v. 1.5):

```
bwa mem -x ont2d HG02982_canu.uncorrected.
fasta reads.joint.fastq | samtools sort -o
reads.joint.mappings.bam -T tmp -
```

The mappings were fed to Nanopolish and corrected in chunks of 50 kb using the helper script "nanopolish_makerange.py" included in the Nanopolish package. Variants were called using "nanopolish variants –consensus" with the optional flag "--min-candidate-frequency 0.1".

```
nanopolish_makerange.py HG02982_canu.
uncorrected.fasta | xargs -i echo nanopolish
variants --consensus selfcorrected.{}.fa -w {}
-r reads.joint.fastq -b reads.joint.mappings.
bam -g HG02982_canu.uncorrected.fasta -t 4
--min-candidate-frequency 0.1 | sh
```

By this means, we corrected 127,801 positions in the initial assembly. The self-corrected assembly was further polished with the Illumina library. To this end, we trimmed the Illumina reads to get rid of any adapters in the sequences using trimgalore (v 3.7, https://github.com/FelixKrueger/TrimGalore).

```
trim_galore --fastqc --paired --retain_
unpaired gzip pair1.fastq pair2.fastq
```

The trimmed reads were mapped with BWA mem (v.0.7.12)[32] in paired-end mode and the mappings converted to a sorted bam files using samtools sort. PCR

duplicates were removed with Picardtools (v. 2.8.2, https://broadinstitute.github.io/picard/).

```
bwa mem HG02982_canu.selfcorrected.fasta
reads.p1.fastq reads.p2.fastq | samtools sort
-o reads.paired.mappings.bam -T tmp -;
java -jar picard.jar MarkDuplicates I=reads.
paired.mappings.bam O=reads.paired.mappings.
markdup.bam M=reads.paired.mappings.markdup.
bam
```

Polishing was performed with Pilon (v 1.22)[19], resulting in 132,336 residual errors being corrected.

```
java -Xmx96G -jar pilon-1.22.jar --threads 12
--genome HG02982_canu.selfcorrected.fasta
--frags reads.paired.mappings.markdup.bam
--output HG02982_canu.selfcorrected.pileon
--outdir pilon_corrections --changes --vcf
--tracks --fix all
```

To run racon (v 1.3.1, see Supplementary Table 4), we mapped the Illumina reads onto the polished reference with bwa, sorted the alignments with samtools and removed duplicates as described above. The resulting alignments were provided to racon as an input:

```
racon -u -t 12 reads.fastq mappings.sam
HG02982_chrY_v1.fasta
```

**Variant calls.** For variant calls, the Illumina data were mapped onto the GRCh38 or the HG02982 assembly, respectively, and processed the same way as detailed above. Variants were called using GATKs Haplotype Caller with the following optional flags: "--genotyping-mode DISCOVERY --sample-ploidy 1".

```
java -jar gatk-package-4.0.0.0-local.jar
HaplotypeCaller -R reference.fa -I mappings.
bam --genotyping-mode DISCOVERY -O variants.
vcf
```

**Repeat annotations.** Repeat annotations were performed using RepeatMasker (v. 4.0.7) with rmblastn v. 2.6.0+ as the engine. To be comparable, the annotations for both the HG02982, as well as the GRCh38 assembly were performed the same way. We used the RepBase-20170127 as the repeatmasker database, and Homo sapiens as the query species. Divergence of the repeat annotations to their consensus was calculated using the "calcDivergenceFromAlign.pl" utility included in the RepeatMasker package.

```
RepeatMasker -e ncbi -pa 12 -s -species human
-no_is -noisy -dir ./outDir -a -gff -u reference.
fa
```

**Whole-genome alignments.** Whole-genome alignments to GRCh38 were produced using last (v. 914) with the following parameters as suggested by the developer for highly similar genomes for indexing and alignments:

```
lastdb -uNEAR -R01 index reference.fa
lastal -e25 -v -q3 -j4 index query.fa >
mappings.maf
```

Single best placements of query sequences were retained using the "last-split" script included in the last alignment package. Alignments were filtered for a maximum mismap probability of 10e–5. The alignments were converted to psl format for further processing.

**Comparison with WGS PacBio data.** The PacBio data from the Ashkenazim Son (Coriel ID NA24385) produced by the genome in a bottle consortium was also assembled using Canu (v. 1.6) using default assembly parameters and assuming a genome size of 3.2 Gb:

```
canu -p NA24385 -d NA24385_canu genomeSize=3.
2g -pacbio-raw data/fastq/*fastq.gz
gridOptionsExecutive='--mem-per-cpu=16g
--cpus-per-task=2'
```

After genome assembly, we performed a whole-genome alignment to GRChg38 and retained single best placements as mentioned above. To identify contigs belonging to the Y chromosome, we performed the following filtering steps: for contigs, which have local best placements on a chromosome different than the Y, we filtered out those whose proportion of mapped bases is higher on a sequence from the reference assembly different from the Y chromosome. Additionally, we filtered out any alignments with a mismap probability higher than 10e–5. By this means, we retained 184 contigs mapping 15,308,468 base pairs on the Y chromosome (see Supplementary Data 6)

**SV calls.** SVs were called with assemblytics[25]. To this end, we produced whole-genome alignments using nucmer from the Mummer package (v. 3.22)[33]. The resulting delta file was passed to assemblytics, with the required unique anchor length set to 10000 bp.

```
nucmer -maxmatch -l 100 -c 500 GRCh38.chrY.fa
HG02982_chrY_v1.fasta -prefix HG02982_vs_HG38
```

```
Assemblytics HG02982_vs_HG38_.delta
HG02982_chrY_v1.vs.hg38_10kanchor.50kmax
10000 bin/Assemblytics/
```

**Read depth duplication detection**. We estimated absolute copy number with a depth of coverage approach using the Illumina data[22]. We masked all common repeats as identified by RepeatMasker (see above) and tandem repeat finder. We created non-overlapping 36-mers of the raw reads, which were mapped onto the assembly using GEM (v 2)[34] allowing for a divergence of up to 5%. The read depth was calculated in non-overlapping windows of 1 kb of non-repetitive sequence. After correcting for GC content using mrCanavar (v. 0.51), we normalized by the mean read depth. To assign a copy number to each gene, we calculated the median copy number of all windows intersecting a gene. For the hg38 Y chromosome, a set of custom single-copy regions needed to be provided to the CN caller as calibration. These regions were inferred by subtracting the reference WGAC (whole-genome assembly comparison, UCSC track genomic superdups) segmental duplication track from the whole Y chromosome and keeping only stretches of single-copy sequence longer than 2 kb.

**Gene annotation**. The annotation of the HG02982 assembly was performed by trying to assign the genes present in the Y-chromosome annotation of GRCh38 gencode version 27. For this purpose, we downloaded the gff3, the transcript sequences and the protein sequences that corresponded to the Y-chromosome annotation and performed transcript and protein mappings with GMAP (v. 20170317[35]) and exonerate (v. 2.2.0[36]), respectively. Additionally, a numeric index was assigned to each gene in the HG38 Y chromosome according to the order in the chromosome. Next, we combined all the data (transcript mappings, protein mappings and gene synteny) with an in-house script (available at https://doi.org/10.6084/m9.figshare.7359065.v1) to locate each gene in our assembly and assign parts of the assembly to their corresponding region in the Y chromosome of GRCh38. After following the strategy mentioned above for all the genes, we took a closer look to the protein-coding genes, by manually checking some of the mappings in order to determine possible errors in the sequence caused by the Nanopore reads that could introduce frameshifts or internal stop codons in the aminoacidic sequence.

**Illumina WGBS sequencing and methylation calls**. Two micrograms of genomic DNA from a lymphoblastoid cell line (HG02982) were spiked with unmethylated bacteriophage λ DNA (5 ng of λ DNA per microgram of genomic DNA; Promega) and with methylated T7 phage DNA (5 ng of T7 DNA per microgram of genomic DNA). The DNA was sheared to 50–500 bp in size using Covaris LE220 ultra-sonicator, and fragments of 150–300 bp were size-selected using AMPure XP beads (Agencourt Bioscience). The libraries were constructed using the KAPA Library Preparation Kit with no PCR Library Amplification/Illumina series (Roche-Kapa Biosystems) together with the NEXTFLEX® Bisulfite-Seq Barcodes (Perkin Elmer). After adaptor ligation, the DNA was treated with sodium bisulfite using the Epi-Tect Bisulfite kit (Qiagen) following the manufacturer's instructions. Enrichment for adaptor-ligated DNA was carried out through seven PCR cycles using KAPA HiFi HotStart Uracil+ReadyMix PCR 2x Kit (Roche-Kapa Biosystems). Library quality was monitored using the Agilent 2100 Bioanalyzer DNA 7500 assay, and the library concentration was estimated using quantitative PCR using the KAPA Library Quantification Kit for Illumina® Platforms, v1.14 (Roche-Kapa Biosystems).

Paired-end DNA sequencing (2×101 bp) of the converted libraries was performed using the HiSeq 2500 (Illumina) following the manufacturer's protocol with HiSeq Control Software (HCS) 2.2.68. Primary data analysis, image analysis, base calling, and quality scoring of the run, was processed using the manufacturer's software Real Time Analysis (RTA 1.18.66.3) and followed by generation of FASTQ sequence files by CASAVA.

We used the gemBS pipeline[37] using the default parameters to perform the analysis. The reference genome used for the alignment was GRCh38. Methylated and unmethylated cytosine conversion rates were determined from spiked-in bacteriophage DNA (fully methylated phage T7 and unmethylated phage lambda). The under and over conversion rates for the sample were <1 and ~ 5%, respectively. Only uniquely mapping reads were retained for downstream analysis. The comparison with the Nanopore calls was performed for all canonical CpG sites on the Y chromosome where there was sequencing data available from both experiments. The comparison took account of the variable precision of the methylation estimates due to variation in sequencing coverage between sites so that low-coverage sites did not affect the comparison.

**Nanopore methylation calls**. The methylation status was called using Nanopolish[10] as suggested by the developers. To this end, we aligned the Nanopore reads to GRCh38 with minimap2[38] and sorted with samtools (v 1.5). The calls were performed in 200 kb windows.

```
minimap2 -a -x map-ont chrY.fa joint_reads.
fastq | samtools sort -T tmp -o joint_reads.
mappings.bam
    samtools index joint_reads.mappings.bam
```

```
nanopolish call-methylation -v --progress -t
8 -r joint_reads.fastq -b joint_reads.
mappings.bam -g chrY.fa -w"chrY:$start-$stop"
> methylation_calls.tsv
```

Finally, we calculated the methylation frequency and log-likelihood ratios of methylation at each position:

```
calculate_methylation_frequency.py -i
methylation_calls.tsv
```

We filtered out any position with <10 reads in either the WGBS or the Nanopore data. Additionally, any position with a log-likelihood ratio of <2.5 in the Nanopore data were also excluded.

**Code availability**. The custom script used for the gene annotation has been deposited at Figshare at https://doi.org/10.6084/m9.figshare.7359065.v1.

## Data availability

All raw sequencing data for this study have been deposited at the European Nucleotide Archive (ENA) under the study accession PRJEB28143. The assembly is deposited at the ENA under the accession ULGL01000000. The WGS assembly for the NA24385 individual, the repeat masker tracks for the GRCh38 chrY and HG02982 assemblies, and the methylation calls from the Illumina WGBS and the Nanopore data are deposited at Figshare at https://doi.org/10.6084/m9.figshare.7358480.v1. The source data underlying Figs. 1a–d and 2a–c are provided as a Source Data File. A Reporting Summary for this Article is available as a Supplementary Information file. All other relevant data are available upon request.

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

## Acknowledgements

This study was supported by the Spanish Ministry of Economy and Competitiveness with Proyectos de I+D "Excelencia" y Proyectos de I+D+I "Retos Investigación" BFU2014-55090-P awarded to T.M.-B. and O.F., Centro de Excelencia Severo Ochoa 2013–2017 and Centro de Excelencia Maria de Maeztu 2016–2019. We acknowledge the support from the CERCA Programme of the Generalitat de Catalunya, institutional support from the Spanish Ministry of Economy, Industry and Competitiveness (MEIC) through the Instituto de Salud Carlos III, from the Generalitat de Catalunya through the Departament de Salut and Departament d'Empresa i Coneixement, and co-financing by the Spanish Ministry of Economy, Industry and Competitiveness (MEIC) with funds from the European Regional Development Fund (ERDF) corresponding to the 2014–2020 Smart Growth Operating Program. L.F.K.K. is supported by an FPI fellowship associated with BFU2014-55090-P (MINECO/FEDER, UE). M.K. is supported by a Deutsche Forschungsgemeinschaft (DFG) fellowship (KU 3467/1-1). T.M.-B. is supported by BFU2017-86471-P (MINECO/FEDER, UE), U01 MH106874 grant, Howard Hughes International Early Career, Obra Social "La Caixa" and Secretaria d'Universitats i Recerca del Departament d'Economia i Coneixement de la Generalitat de Catalunya. D.J. is supported by a Juan de la Cierva fellowship (FJCI-2016-29558) from MICINN.

## Author contributions

T.M.-B. conceived the study, L.F.K.K., J.G.-G., A.S.A., M.K., S.H., D.J., and T.A. performed computational analysis, E.L. developed and performed the purification protocol, E.J. and O.F. cultured cells and performed the flow cytometry, R.A.A., M.A.-E., M.G., I. G., and M.H.S. prepared materials and/or performed the sequencing. L.F.K.K., E.L., and T.M.-B. wrote the manuscript with input from all authors. All authors approved the manuscript.

## Additional information

**Competing interests:** The authors declare no competing interests.

