## [Peer Review File · Nature Communications]

Reviewers' comments:

Reviewer #1 (Remarks to the Author):

The authors present a very nice work flow for sequencing the Y chromosome using MinION sequencing combined with flow sorting of chromosomal DNA to isolate the Y chromosome. The authors present an assembly of the first Y chromosome of African origin. The combination of chromosome flow sorting with Nanopore is an excellent approach. However, to my knowledge it has proven challenging to many users and it is to their credit that this team have managed to succeed with this approach. I broadly support the publication of this paper although I believe it can (and perhaps should) be significantly improved. I believe it is important that the computational pipeline used for handling these data be corrected as this method will no doubt be used by other groups in the future.

Major points.

The authors applied Nanopolish to their data sets in order to determine SNPs and methylation status. This is to their credit. Unfortunately I believe they have not used the correct component of Nanopolish for SNP calling. Instead of using the Nanopolish command: `nanopolish variants --consensus` they should have used: `nanopolish variants` i.e there was no need to correct a consensus - rather they should have called SNPs with respect to a reference. Using SNP calling rather than consensus calling will likely be far more informative.

The authors may find that applying both Pilon and Racon will further improve their analysis. Pilon alone appears to exhibit some unusual edge cases where errors occur within close proximity to one another.

I take issue with the description of "well correlated" for a Pearson's r of 0.502-0.583. There is at best moderate correlation. Looking at the supplementary figures (20-23) doesn't increase my confidence here. I think it is important to not over-state these observations.

I am unable to follow what is being shown in Supplementary Figure 6. I suspect something is missing from the plot, but if not, I would encourage the authors to revisit this plot from a naive readers perspective!

Minor points.

Figure 1B is somewhat confusing as the text suggests 11-fold enrichment whereas the plot shows just greater than 30x coverage. Perhaps plotting the data on a log scale would make clearer that the background coverage is $< < 1$?

Figure 1C/D strictly the Read N50 measurement is in bases, not base pairs. It might be helpful to the unobservant casual reader to clearly indicate the use of a log scale in panel C.

The authors note that they re-basecalled the data using Albacore v2.1 for use with Nanopolish. It is less clear which version of the base caller was used for the reads in the canu assembly step. If the authors were using older versions of the base caller then I would strongly encourage them to use the latest available base caller version for the canu step (or clarify if this was used).

Although I appreciate that the recovery from chromosome sorting was likely low, I would encourage the authors to use more DNA in their rapid sequencing preps in future. Loading only 200ng is likely to result in the relatively low yields seen here!

Reviewer #2 (Remarks to the Author):

In Kuderna et. al. the authors address one of the major challenges in genome assembly: the Y-chromosome, by utilizing the long reads provided by nanopore sequencing on flow sorted Y-

chromosomes.

Overall the manuscript is very well written, includes necessary QC assessments, and presents an important advance in genome assembly of difficult regions. My only major criticism is regarding the underdeveloped section on DNA methylation (below), though the manuscript would be a complete story even without that section. Also the authors should note in the final paragraph that one of the limitations of their method is that sorting individual chromosomes can be quite challenging (this reviewer has tried with only limited success). It is also limited with respect to sortable chromosomes, ie. Of a certain size and distinguishable from other chromosomes.

Comments:

Line 81: perhaps state the amount sequenced as “over 2.3 Gbp”

Line 83: Add in the numbers, ie “at the lower end of the spectrum” being what? And what were your numbers? I know they are in the figure, but it would be nice to have it in the text as well.

Lines 117-119: The wording of this sentence is awkward – maybe change to “Due to the sharp coincidence with the PAR-1 boundary, we do not suspect that the reduced coverage is a technical artifact of library preparation.”

Lines 140-143: This should be stated differently. It currently reads as though the 4 genes were partially missing, but then it says they are a technical issue. It should be phrased where it is noting that it is a technical issue up front. Eg “We also note that four of these genes (list) were partially missing from our assembly due to technical challenges in the PAR-1 region”

Methylation portion: This section is very underdeveloped. I understand wanting to include it, but the analysis is very cursory. Further, the analysis of the bisulfite data alone is challenging for the Y chromosome. I would suggest either removing it entirely (as the work stands on its own without it), or showing at least: methylation levels over genic regions normalized to gene length (ie. Methylation +5000 bp before TSS, % through the gene body, and then 5000 bp after TES), showing numbers on the %CG methylated and %CH methylated, methylation through CG islands, comparing methylation in regions with corresponding gene expression, or known biology of whether the genes should be expressed in LCLs, and absolutely include a description of the methods in the supplement which was lacking entirely. More description of the power to call the sites is also important... ie how did nanopolish perform? How many C's could be assessed for methylation status? Etc...

Line 184: this is interesting – it certainly makes sense that residual dyes would interfere with PacBio sequencing...

Figure comments:

Figure 1A&B: beautiful sorting! Wow!

Figure 1D: one idea would be to have the figure as a scatterplot w/ the y-axis as the yield of the run and the x-axis as the N50 read length? Just a suggestion.

Figure 2B: What does the color correspond to?

Supp. Fig. 6: this figure is hard to interpret – what are the coverage lines? Is it just the black vertical lines? Also indicate with an arrow or something the PAR-1 dropoff.

Supp. Fig. 8: This is really cool – possibly worth being in main figure 2?

Reviewer #3 (Remarks to the Author):

The authors in this study have sequenced and produced a reference-grade assembly of a Y chromosome, from an individual from Africa. The Y chromosome was enriched by flow sorting and the sequencing was performed on the nanopore sequencing platform – giving the advantage of both long read and methylation in the signal file. Overall, I felt the manuscript did a nice job of stating the advance in assembly contiguity of the current human GRCh38 status. However, I broadly felt like the manuscript would benefit from adding details of any new findings that are outside of what was previously known. Although flow sorting as an enrichment approach is a clever idea, it does require a level of specialization in both equipment and training. Further, as long reads from PacBio/ONT increase the price decreases it may be difficult to justify this approach with the current read lengths and throughput. I recognize that this is the challenge of any technology paper (as it is a moving target). The writing and presentation of the manuscript was clear and well organized. Comments below are provided to help improve and expand on interesting findings present in the existing manuscript:

1. I acknowledge the achievement of assembling the Y chromosome to “reference grade” quality, however it is distracting when the authors state that they have reached complete chromosome-scale assembly. This is absolutely not true – the authors have not completed the highly repetitive q-arm. Further, with the current strategy they are unable to reach the milestone of completing another human centromere assembly. The authors need to update the text throughout the manuscript that reference completeness: specifically the abstract (L42-44).
2. The authors have spent the majority of the manuscript describing the assembly and comparison analysis to what is ‘known’. That is, direct assembly comparisons were performed against known ampliconic sequences and structural variants. It would improve the paper and emphasize the importance of their work if the authors also discussed any novel findings or assembly predictions that are not currently part of GRCh38.
3. The ability to study methylation patterns on the Y chromosome is one of the most interesting features of this manuscript. Readers may appreciate more information about perceived challenges with methylation and nanopore base calling. Also, it would be useful to include any additional interesting biological information about methylation that the authors gained from including these data –other than the correlation with ~matched ENCODE data.
4. The authors should clearly label the centromere gap on their annotated dot plot in Figure 2. Otherwise it may give the casual reader the impression that they closed that particular gap.
5. Also in figure 2, the gorilla treemap may be best placed in supplemental. It is not clear if the alignment/tree plot is due to assembly or the natural organization of the chromosome and does not seem central to the analysis.
6. In table 1 the authors are mapping heterochromatin with sequence identity of 99.69%. It was not clear if the authors are also including heterochromatic satellite DNAs in this estimate, and if so it is likely that they are not characterizing the entirety of the dataset – due expected inherent variation across the q-arm and centromere. It may be useful to directly state how much of the chrY sequence

data remains uncharacterized and/or unassembled.

7. It would be useful to include statement about the inability to perform ultra long (UL)-protocol, even with 9M flow sorted chromosomes (due to the requirements of starting material 10ug vs 100ng; and expected decreased throughput). The goal of completing a human chromosome with the read lengths presented in figure 1 will not match the expectation of the reader (ie. why not complete with UL-reads).

8. In addition to comparisons about contiguity, it may be valuable to provide an sohort insertion/deletion comparison (similar to that shown on Figure 1c) with the analysis presented here with the PacBio GIAB datasets.

Reviewer #4 (Remarks to the Author):

Kuderna et al. combine flow sorting with Oxford Nanopore (ONT) long read sequencing to develop an approach to selectively sequence and assemble human Y chromosomes. They apply their technique to cell line HG02982, a sample of African origin that was part of the 1000 Genomes project. While it seems likely that we will see many more whole genome assemblies from long reads the near future (including on male samples), I agree with the authors that selective, cost-efficient sequencing and assembly of the Y chromosome can be valuable tool for future studies in this domain. In general the results appear sound, the computational pipeline follows standard practices, and the paper is well written. It could potentially be suitable for Nature Communications, given that the specific comments below are addressed:

- To show that their strategy is superior to WGS, the authors perform an assembly of NA24385 (Ashkenazi son from genome in a bottle). As far as I remember, that data is comparatively old and has much shorter reads than present-day PacBio data. Could the authors please include read length statistics for these data? To get closer to an apples-to-apples comparison, it might make sense to, for instance, to include recent ONT data for the same sample (ftp://ftp-trace.ncbi.nlm.nih.gov/giab/ftp/data/AshkenazimTrio/HG002_NA24385_son/Ultralong_OxfordNanopore/), although this data still seems to be low coverage according to the README (with more sequencing planned), so probably it'd need to be combined with the PacBio data. Another public source of ONT data of a male sample would be the "cliveome" (<https://github.com/nanoporetech/ONT-HG1>). The authors cite Tomaszewicz et al. (2016) for an Gorilla Y chromosome assembly. In this context, it might be worth noting that the chimpanzee assembled as part of the recent Kronenberg et al. paper (Science, 2018) was male, so it would be interesting to compare to that assembly. While it makes sense to compare to existing assemblies, I would be cautious in claiming that the assemblies resulting from the strategy developed in this paper are in general superior to WGS assemblies. In my view, the main advantage lies in the cost-effectiveness for Y chromosome studies.

- P7,152: I assume this "read depth" approach refers to the Illumina-data-based CNV calling described in Methods. Is this increased copy number reflected in the assemblies? If that should not be the case, could the authors comment on the (potential) reasons. If the assemblies show multiple copies consistent with the CNV calls (and hence reveal the precise architecture), that would be a nice addition to the paper.

- The data access section still contains place holders, so I could look at the data.

- P7: "We manually validate all structural variants called in HG02982 in the 1000 Genomes Project in our assembly (see supplementary material)." In the supplement, I see three IGV screen shots. Can I conclude from that that 1KG made three SV calls? I think the authors should make more explicit what they mean by "validating" a call. In all three cases, Illumina data, 1000G calls, and calls made from the genome assembly do not seem to be in very good agreement. Working out the reasons for these differences would be important to develop a better understanding of the strengths/limitations of the proposed approach.

- The fact that methylation status can be inferred due to using native DNA increases the appeal of this approach. However, the analysis of methylation levels is somewhat shallow and the reported correlation levels. In my view, some sort of control experiment would be needed to establish the reliability of the methylation calls.

- The paper is mostly technological and does not include much biological interpretation of the assembled sequence.

- The ONT technology comes with the option of "read until", i.e. of using the sequencing device itself to selectively eject unwanted molecules from the nanopores, thereby enriching for particular sequences (e.g. doi: 10.1038/nmeth.3930). I wonder whether the authors can comment on whether they considered to use this feature instead of the flow sorting and share their view on the feasibility.

- In Supp. Fig. 6, I do not see a coverage curve. Presumably something went wrong here with embedding the figure in the document? For this reason, I could not assess this drop in coverage at the PAR-1 boundary mentioned in the text. Could the missing genes (P6,L140) be due to the reduced coverage? In general, the coverage levels are at the low end of what one would want for genome assembly (and generation of a consensus), especially from noisy ONT data. I suspect that the assembly quality would go up quite a bit at an increased coverage.

- Figure 2: Panel 2: it is hard to judge whether the breaks in the dot plot coincide with the region boundaries. I suggest to color the background of the dot plot according to the region. Panel C: Colors for Insertion and Repeat_contraction are hard to distinguish (at least in my printout). Please add axis labels.

- P24,L472ff. This is duplicate and appears again on Page 26.

- P29,L585: likelyhood -> likelihood

Reviewer #1 (Remarks to the Author):

The authors present a very nice work flow for sequencing the Y chromosome using MinION sequencing combined with flow sorting of chromosomal DNA to isolate the Y chromosome. The authors present an assembly of the first Y chromosome of African origin. The combination of chromosome flow sorting with Nanopore is an excellent approach. However, to my knowledge it has proven challenging to many users and it is to their credit that this team have managed to succeed with this approach. I broadly support the publication of this paper although I believe it can (and perhaps should) be significantly improved. I believe it is important that the computational pipeline used for handling these data be corrected as this method will no doubt be used by other groups in the future.

Major points.

The authors applied Nanopolish to their data sets in order to determine SNPs and methylation status. This is to their credit. Unfortunately I believe they have not used the correct component of Nanopolish for SNP calling. Instead of using the Nanopolish command: `nanopolish variants --consensus` they should have used: `nanopolish variants` i.e. there was no need to correct a consensus - rather they should have called SNPs with respect to a reference. Using SNP calling rather than consensus calling will likely be far more informative. The authors may find that applying both Pilon and Racon will further improve their analysis. Pilon alone appears to exhibit some unusual edge cases where errors occur within close proximity to one another.

We thank the reviewer for these suggestions, as it is in our prime interest to lower the Nanopore-derived error rate within our assembly to enable more fine-scale analysis. We re-ran the latest version of Nanopolish (v. 0.10.1) in variant mode only, without consensus calling. Additionally, we applied increasing cut-offs from (0-0.9 in increments of 0.1) of the support fraction flag, i.e. the proportion of reads supporting a given variant. We then incorporated these changes into the assembly and realigned the differently polished assemblies to the GRCh38 Y chromosome to calculate the percentage of identical bases. The alignments were performed with last (see Methods) and identity was defined as either matches/(matches + mismatches) or matches/(matches + mismatches + deleted bases + inserted bases).

We find little difference (0.065%) between polishing with consensus mode and variant calling only without any filtering. Overall, the consensus mode achieves a slightly higher % identity (98.803% vs. 98.868%). We also compared the total number of matches, mismatches, insertions and deletions. While the consensus mode does have a slightly higher ab initio identity, and fewer deletions, we find the variant mode to increase the number of matches by around 200 Kbp. Notwithstanding we find the average identity of the assembly to slightly decrease. Given these tradeoffs, we believe that the different polishing strategies should be discussed, but we don't think it is clear that using variant-calling only significantly improves the assembly over the consensus mode. We have included these analyses in the supplementary material at supplementary Table S4.

The support-fraction cutoffs don't have much impact on the assembly's quality. At best, a small (0.1) cutoff slightly improves the error rate, but it seems that beyond that a lot of true positive error-corrections get filtered out too.

We additionally ran one round of Racon on our assembly. We find this to slightly decrease the identity to GRCh38 by 0.124% compared to the Pilon-only corrected assembly. However, polishing with Racon also allows us to additionally align 143 Kb of matches, corrected 116 Kb of deletions and 2 Kb of insertions, although it also introduced 6 Kb of additional mismatches.

We used Nanopolish's consensus mode as this is the suggested mode by the developers to polish a genome assembly. It is not completely clear to us if the Reviewer is suggesting we should have used the Nanopore data to just call variants using the available reference instead of producing a de-novo assembly. If so, we disagree, as there will be variation beyond SNV that are not straightforward to capture using re-sequencing but is easily incorporated into the assembly.

We have adapted the manuscript to reflect these suggestions accordingly.

Matches	Mismatches	Insertions	Deletions	perc. Identity	perc. Identity SNP only	assembly
20242759	30248	12633	202425	0.988027	0.998508	NP_variants_min_0
20243039	30301	12701	202508	0.988017	0.998505	NP_variants_min_0.1
20234195	30500	12886	203878	0.987927	0.998495	NP_variants_min_0.2
20206603	30969	13277	208557	0.987644	0.99847	NP_variants_min_0.3
20251615	31840	13838	219506	0.987075	0.99843	NP_variants_min_0.4
20198119	32856	14191	238764	0.986047	0.998376	NP_variants_min_0.5
20036473	33449	14533	263428	0.984696	0.998333	NP_variants_min_0.6
19969862	34413	14867	288018	0.98339	0.99828	NP_variants_min_0.7
19944500	35090	15062	299929	0.98275	0.998244	NP_variants_min_0.8
19908562	35372	15195	304139	0.982495	0.998226	NP_variants_min_0.9
20125302	29331	13094	188110	0.988675	0.998545	NP_consensus
20459952	24716	15154	27771	0.996705	0.998793	NP_consensus_pilon
20602540	50440	8321	20051	0.996189	0.997558	NP_consensus_pilon_racon

I take issue with the description of “well correlated” for a Pearson’s r of 0.502-0.583. There is at best moderate correlation. Looking at the supplementary figures (20-23) doesn’t increase my confidence here. I think it is important to not over-state these observations. □

We have reworked the methylation section completely and removed this sentence as a consequence. For a direct comparison, we have produced novel dataset of Illumina whole genome bisulfite sequencing of the same cell line, and find a Pearson’s r of 0.82. We claim this value is ‘good concordance’ rather than ‘well correlated’ and discuss potential reason for remaining differences.

I am unable to follow what is being shown in Supplementary Figure 6. I suspect something is missing from the plot, but if not, I would encourage the authors to revisit this plot from a naive readers perspective!

We apologize for this inconvenience. There has been a rendering issue with this plot on some platforms, in which the data points were removed from the plot in the manuscript. We have replaced it with a correctly rendered version.

Minor points.

Figure 1B is somewhat confusing as the text suggests 11-fold enrichment whereas the plot shows just greater than 30x coverage. Perhaps plotting the data on a log scale would make clearer that the background coverage is $\ll 1$?

The enrichment factor was calculated over a random sampling from a diploid male human genome as the background, to compare the selectivity of flow sorting to WG sequencing (see supplementary material). In our sequencing data, we encounter Y chromosomal sequences 110-times more often than we would expect to encounter them in a WG sequencing experiment, without accounting for sequencing biases.

Figure 1C/D strictly the Read N50 measurement is in bases, not base pairs. It might be helpful to the unobservant casual reader to clearly indicate the use of a log scale in panel C.

We have changed the axis annotation to state bases instead of base pairs. We now explicitly state the use of a log-scale in the figure legend.

The authors note that they re-basercalled the data using Albacore v2.1 for use with Nanopolish. It is less clear which version of the base caller was used for the reads in the canu assembly step. If the authors were using older versions of the base caller then I would strongly encourage them to use the latest available base caller version for the canu step (or clarify if this was used).

We now explicitly state what algorithm was used for the basecalling used in the assembly (MinKNOW 1.7.10 using Albacore 1.1). This was the default basecaller available by the time of data production. While we agree that ideally all data would have been re-called using the latest version now available, the choice of basecalling algorithms is a moving target and all analysis are based on this assembly.

Although I appreciate that the recovery from chromosome sorting was likely low, I would encourage the authors to use more DNA in their rapid sequencing preps in future. Loading only 200ng is likely to result in the relatively low yields seen here!

We appreciate this comment and are aware that the low amount of loaded DNA is likely to be the cause for the low yields. At the time the sequencing experiments started, the recommended loading amount by ONT for rapid kits was 200 ng, which was subsequently increased to 400 ng. The reason we kept loading 200 ng was indeed the low recovery from flow sorted chromosomes, and the fact that our calculations of required number of sorted chromosomes were based off the initial yields from the initial test-run loading that amount. Given the difficulties in quantification and recovery of flow sorting material, we did not want to change these conditions.

Reviewer #2 (Remarks to the Author):

In Kuderna et. al. the authors address one of the major challenges in genome assembly: the Y-chromosome, by utilizing the long reads provided by nanopore sequencing on flow sorted Y-chromosomes.

Overall the manuscript is very well written, includes necessary QC assessments, and presents an important advance in genome assembly of difficult regions. My only major criticism is regarding the underdeveloped section on DNA methylation (below), though the manuscript would be a complete story even without that section. Also the authors should note in the final paragraph that one of the limitations of their method is that sorting individual chromosomes can be quite challenging (this reviewer has tried with only limited success). It is also limited with respect to sortable chromosomes, ie. Of a certain size and distinguishable from other chromosomes.

We have added a sentence to the final paragraph stating the challenges of flow-sorting single chromosomes. We also now mention that chromosomes need to be sufficiently different for sorting to be possible.

Comments:

Line 81: perhaps state the amount sequenced as "over 2.3 Gbp"

We have changed the sentence to state 'over 2.3 Gbp'

Line 83: Add in the numbers, ie "at the lower end of the spectrum" being what? And what were your numbers? I know they are in the figure, but it would be nice to have it in the text as well.

We have added the range of sequence yields reported in Jain et al, 2018 as comparison points to our runs, and explicitly state the ranges of N50 values in our data. Additional statistics for each run can be found in the supplementary material.

Lines 117-119: The wording of this sentence is awkward – maybe change to "Due to the sharp coincidence with the PAR-1 boundary, we do not suspect that the reduced coverage is a technical artifact of library preparation."

We have changed to wording the reviewer's suggestion.

Lines 140-143: This should be stated differently. It currently reads as though the 4 genes were partially missing, but then it says they are a technical issue. It should be phrased where it is noting that it is a technical issue up front. Eg "We also note that four of these genes (list) were partially missing from our assembly due to technical challenges in the PAR-1 region"

We have rephrased this sentence and the following one to make it clear that the deletion is artifactual ("We also note that four genes (*ASMTL*, *IL3R*, *P2RY*, *SLC25*) from a comparatively short syntenic block of around 200 kb are partially missing from our assembly due to the aforementioned technical challenges in the PAR-1 region. Mapping the raw data onto GRCh38 shows that this is an artifact, presumably due to insufficient coverage in this region.")

Methylation portion: This section is very underdeveloped. I understand wanting to include it, but the analysis is very cursory. Further, the analysis of the bisulfite data alone is challenging for the Y chromosome. I would suggest either removing it entirely (as the work stands on its own without it), or showing at least: methylation levels over genic regions normalized to gene length (ie. Methylation +5000 bp before TSS, % through the gene body, and then 5000 bp after TES), showing numbers on the %CG methylated and %CH methylated, methylation through CG islands, comparing methylation in regions with corresponding gene expression, or known biology of whether the genes should be expressed in LCLs, and absolutely include a description of the methods in the supplement which was lacking entirely. More description of the power to call the sites is also important... ie how did nanopolish perform? How many C's could be assessed for methylation status? Etc... q

We thank the reviewer for these suggestions and have now completely reworked the methylation section accordingly. We have produced Illumina whole genome bisulfite sequencing data of the same cell line, to have a direct comparison point between the different platforms. We find the WGBS and Nanopolish calls to be in good agreement with each other with a Pearson's r of 0.82.

Minimum Coverage (fold)	Calls Nanopore	Calls Illumina WGBS	Percentage of all CpG Nanopore	Percentage of all CpG Illumina WGBS
1	172,278	84,610	76.26	37.45
5	156,204	38,903	69.14	17.22
10	121,392	5,818	53.73	2.58

We additionally expanded on the methylation analysis of the Nanopore data alone, and by this hope to answer several of the reviewer's questions. We provide figures on the CpG methylation status of Islands, Shores, Shelves and all other CpG. We also discuss the advantages of calling the methylation status with the flow sorted Nanopore data, as we are able to reach into several regions that are largely inaccessible to Illumina WGBS data, such as the Pseudo-autosomal, the X-transposed and the Ampliconic regions. Lastly, we calculated the methylation frequency in protein coding genes in different sequence classes. In accordance with their broad expression patterns, we find transcription start sites of genes in the X-degenerate, the X-transposed and the Pseudoautosomal regions exhibit low

methylation levels. Conversely, we find the transcription start sites of genes in the Ampliconic regions to show high degrees of methylation, consistent with their testis specific expression patterns.

Lastly: Nanopolish is trained exclusively to predict the methylation status of CpG sites. For that reason we are not able to provide information on %CH methylation. The methods for both Nanopolish and WGBS data can be found in the methods section of the main manuscript.

Line 184: this is interesting – it certainly makes sense that residual dyes would interfere with PacBio sequencing...

Indeed, our efforts to produce PacBio sequencing data were to no avail.

Figure comments:

Figure 1A&B: beautiful sorting! Wow!

We appreciate the enthusiasm.

Figure 1D: one idea would be to have the figure as a scatterplot w/ the y-axis as the yield of the run and the x-axis as the N50 read length? Just a suggestion.

The suggested plot is provided below. We nevertheless prefer to keep the panel 1D as is.

Figure 2B: What does the color correspond to?

The coloring of each box is a visual aid to be able to distinguish their delimitations better. We have updated the figure legend to state this explicitly.

Supp. Fig. 6: this figure is hard to interpret – what are the coverage lines? Is it just the black vertical lines? Also indicate with an arrow or something the PAR-1 dropoff.

We apologize for this inconvenience. There has been a rendering issue with this plot on some platforms, in which the data points were removed from the plot in the manuscript. We have replaced it with a correctly rendered version.

Supp. Fig. 8: This is really cool – possibly worth being in main figure 2?

We have included this figure as panel 2C in the main manuscript.

Reviewer #3 (Remarks to the Author):

The authors in this study have sequenced and produced a reference-grade assembly of a Y chromosome, from an individual from Africa. The Y chromosome was enriched by flow sorting and the sequencing was performed on the nanopore sequencing platform – giving the advantage of both long read and methylation in the signal file. Overall, I felt the manuscript did a nice job of stating the advance in assembly contiguity of the current human GRCh38 status. However, I broadly felt like the manuscript would benefit from adding details of any new findings that are outside of what was previously known. Although flow sorting as an enrichment approach is a clever idea, it does require a level of specialization in both equipment and training. Further, as long reads from PacBio/ONT increase the price decreases it may be difficult to justify this approach with the current read lengths and throughput. I recognize that this is the challenge of any technology paper (as it is a moving target).

The writing and presentation of the manuscript was clear and well organized. Comments below are provided to help improve and expand on interesting findings present in the existing manuscript:

1. I acknowledge the achievement of assembling the Y chromosome to “reference grade” quality, however it is distracting when the authors state that they have reached complete chromosome-scale assembly. This is absolutely not true – the authors have not completed the highly repetitive q-arm. Further, with the current strategy they are unable to reach the milestone of completing another human centromere assembly. The authors need to update the text throughout the manuscript that reference completeness: specifically the abstract (L42-44).

Our references to completeness referred to the resolved sequence space available in the public reference genome (GRCh38), which we acknowledge it is not completed. We understand that this might be confusing to some readers regarding remaining unresolved sequences. We have changed the phrasing throughout the manuscript wherever we felt our references to completeness might be misleading, specifically L42-44; L91; L188-190 (line numbers of the original submission). We now also explicitly state the amount of remaining unresolved sequence in GRCh38 in the legend of Table 1.

2. The authors have spent the majority of the manuscript describing the assembly and comparison analysis to what is ‘known’. That is, direct assembly comparisons were performed against known ampliconic sequences and structural variants. It would improve the paper and emphasize the importance of their work if the authors also discussed any novel findings or assembly predictions that are not currently part of GRCh38.

Our manuscript focuses on the technical challenges of Y chromosome assemblies we tackle, and it is for that reason we have chosen human, a species with a very well characterized Y chromosomes, as a benchmark. We have intentionally kept it as a technical workflow description of our method to assemble flow-sorted chromosomes with Nanopore data.

3. The ability to study methylation patterns on the Y chromosome is one of the most interesting features of this manuscript. Readers may appreciate more information about perceived challenges with methylation and nanopore base calling. Also, it would be useful to include any additional interesting biological information about methylation that the authors gained from including these data –other than the correlation with ~matched ENCODE data.

We appreciate this comment from the reviewer, and as a consequence have completely reworked the methylation section of our manuscript. We produced Illumina whole genome bisulfite sequencing for the same individual to assess the concordance of the methylation calls with the current ‘gold standard’, and find a correlation with a Pearson’s r of 0.82.

Minimum Coverage (fold)	Calls Nanopore	Calls Illumina WGBS	Percentage of all CpG Nanopore	Percentage of all CpG Illumina WGBS
1	172,278	84,610	76.26	37.45
5	156,204	38,903	69.14	17.22
10	121,392	5,818	53.73	2.58

We now discuss the advantages using flow sorted nanopore data to address the methylation status on the Y chromosome, as it enables interrogating regions that are inaccessible to Illumina and/or whole genome shotgun sequencing, namely the X-transposed, the Pseudo-autosomal and to some degree the Ampliconic regions. We also assess the methylation status of genic regions within the different sequence classes of the Y chromosome, and find transcription start sites (TSS) of genes within the X-degenerate, the X-transposed and the pseudo-autosomal regions to exhibit low methylation levels, consistent with their broad expression patterns across the body. Conversely, our results suggest high methylation levels in TSS of ampliconic genes, consistent with their specific expression in testis and silencing in the lymphoblastoid cell line used in this project.

4. The authors should clearly label the centromere gap on their annotated dot plot in Figure 2. Otherwise it may give the casual reader the impression that they closed that particular gap.

All gaps in the MSY regions that are currently present in GRCh38 chrY, including centromeric ones, are annotated at the top of Figure 2A. We have additionally introduced a background coloring that makes the breakpoints of the dotplot more obvious, and shows that heterochromatic regions surrounding the centromere are poorly resolved our assembly.

5. Also in figure 2, the gorilla treemap may be best placed in supplemental. It is not clear if the alignment/tree plot is due to assembly or the natural organization of the chromosome and does not seem central to the analysis.

We respectfully disagree with the reviewer on this point. While it is possible that the (unknown) underlying natural organization of the chromosome could be a confounding factor, we nevertheless believe that the figure makes a strong statement about the necessity of developing methods which elucidate this organization more clearly, and the improvement our approach constitutes in this context.

The treemap does not contain any information about alignments, but merely visualizes contig sizes in the different assembly.

6. In table 1 the authors are mapping heterochromatin with sequence identity of 99.69%. It was not clear if the authors are also including heterochromatic satellite DNAs in this estimate, and if so it is likely that they are not characterizing the entirety of the dataset – due expected inherent variation across the q-arm and centromere. It may be useful to directly state how much of the chrY sequence data remains uncharacterized and/or unassembled.

The sequence classes are used as defined in Skaletsky et al, 2003, and the specific sequences coordinates in GRCh38 are included in the Supplementary Table S9. Of the heterochromatic sequences that are resolved in

GRCh38, we recover only around 32.7 %, as stated in Table 1. We do not include any heterochromatic sequences that are not currently part of GRCh38 in our analysis, including the q-arm. We have changed the legend of Table 1 to state this explicitly.

7. It would be useful to include statement about the inability to perform ultra long (UL)-protocol, even with 9M flow sorted chromosomes (due to the requirements of starting material 10ug vs 100ng; and expected decreased throughput). The goal of completing a human chromosome with the read lengths presented in figure 1 will not match the expectation of the reader (ie. why not complete with UL-reads).

We have included a statement about the difficulties to perform the UL protocol from our material (*'Notwithstanding, some challenges to obtain ultra-long reads with from flow sorted chromosomes are still to be overcome, as sorting sufficient material for this protocol is a substantial endeavor.'*). We would like to stress that this need not necessarily be an inherent limitation of flow sorted material, as it might be possible to adapt the protocol given a sufficient quantity of DNA.

8. In addition to comparisons about contiguity, it may be valuable to provide a short insertion/deletion comparison (similar to that shown on Figure 1c) with the analysis presented here with the PacBio GIAB datasets.

We provide an overview of these calls for the reviewer below. We find less events >10bp than from the HG02982 individual, and an excess of events in the 1-10 bp range. The latter is attributable to the lack of polishing of this assembly, which results in a high degree of remaining indels from the Pacbio data. Our intention was to show the limitations in reconstructing certain sequence classes from Y chromosome assemblies from whole genome shotgun data, rather than bringing the Ashkenazim son assembly to the highest possible quality. Given the biases in the nanopore data (which are discussed in the main manuscript), and the missing sequences from the whole genome shotgun assembly, we believe an apple-to-apples comparison of the two call sets is not possible, and prefer not to include it in the manuscript. Additionally, we have decided to move figure 1C to the supplementary material.

Insertion		
Size range	Count	Total bp
1-10 bp	9984	12227
10-50 bp	86	1340
50-500 bp	17	4039
500-10000 bp	4	5937
Total	10091	23543
Deletion		
Size range	Count	Total bp
1-10 bp	16608	19694
10-50 bp	152	2688
50-500 bp	5	453
500-10000 bp	0	0
Total	16765	22835
Tandem_expansion		
Size range	Count	Total bp
1-10 bp	0	0
10-50 bp	0	0
50-500 bp	2	558
500-10000 bp	1	2241
Total	3	2799
Tandem_contraction		
Size range	Count	Total bp
1-10 bp	0	0
10-50 bp	0	0
50-500 bp	1	171
500-10000 bp	0	0
Total	1	171
Repeat_expansion		
Size range	Count	Total bp
1-10 bp	0	0
10-50 bp	0	0
50-500 bp	1	319
500-10000 bp	0	0

Total	1	319
Repeat_contraction		
Size range	Count	Total bp
1-10 bp	0	0
10-50 bp	0	0
50-500 bp	5	1038
500-10000 bp	4	9505
Total	9	10543
Total for all variants	26870	60210 bp

Reviewer #4 (Remarks to the Author):

Kuderna et al. combine flow sorting with Oxford Nanopore (ONT) long read sequencing to develop an approach to selectively sequence and assemble human Y chromosomes. They apply their technique to cell line HG02982, a sample of African origin that was part of the 1000 Genomes project. While it seems likely that we will see many more whole genome assemblies from long reads the near future (including on male samples), I agree with the authors that selective, cost-efficient sequencing and assembly of the Y chromosome can be a valuable tool for future studies in this domain. In general the results appear sound, the computational pipeline follows standard practices, and the paper is well written. It could potentially be suitable for Nature Communications, given that the specific comments below are addressed:

- To show that their strategy is superior to WGS, the authors perform an assembly of NA24385 (Ashkenazi son from genome in a bottle). As far as I remember, that data is comparatively old and has much shorter reads than present-day PacBio data. Could the authors please include read length statistics for these data?

The read N50 of the data used to produce the assembly is 15.8 kb (mean 11.4 kb, median 8.3 kb). The range of read N50 of the five recently published great ape assemblies by Kronenberg et al (2018) range from 13.4 kb – 18.6 kb. Although we acknowledge that somewhat longer read lengths are possible to achieve with PacBio, we believe that the read length distribution of the NA24385 data is representative of the current state of the art and thus, in our opinion, an adequate test case.

To get closer to an apples-to-apples comparison, it might make sense to, for instance, to include recent ONT data for the same sample (ftp://ftp-trace.ncbi.nlm.nih.gov/ftp/data/AshkenazimTrio/HG002_NA24385_son/Ultralong_OxfordNanopore/), although this data still seems to be low coverage according to the README (with more sequencing planned), so probably it'd need to be combined with the PacBio data.

It is our opinion that by assembling the PacBio data only we have provided sufficient evidence of the limitations of whole genome shotgun assemblies in the context of the Y chromosome and do not think that combining all different datasets available for this individual will get us closer to an apples to apples comparison. Specifically, given the similar overall read-length distribution between the PacBio NA24385 WGS data and our flow sorted chromosomes (N50 of 15.8Kb vs 18.7 Kb) the differences in sequence content is likely to better reflect the limitations that the genome architecture imposes upon the assembly.

Another public source of ONT data of a male sample would be the "cliveome" (<https://github.com/nanoporetech/ONT-HG1>).

While the 'cliveome' is publicly available, it is subject to restrictive copyright licensing which complicates its use in this context. For this reason, we have decided not to include it in our work.

The authors cite Tomaszewicz et al. (2016) for an Gorilla Y chromosome assembly. In this context, it might be worth noting that the chimpanzee assembled as part of the recent Kronenberg et al. paper (Science, 2018) was male, so it would be interesting to compare to that assembly.

We have included the comparison Tomaszewicz et al. (2016) solely due to the fact that it is based on a similar methodology, upon which we improve. However, we have performed a comparison to Y chromosomal sequences recovered from a whole genome shotgun assembly from NA24385 (Human, Ashkenazi son), which constitutes a similar case to the Chimp.

While it makes sense to compare to existing assemblies, I would be cautious in claiming that the assemblies resulting from the strategy developed in this paper are in general superior to WGS assemblies. In my view, the main advantage lies in the cost-effectiveness for Y chromosome studies.

We have rephrased our conclusion statement regarding sequence class completeness to state that the comparison to WGS only holds for this project and the current state of the art (*"We show that we not only outperform previous efforts that sought to achieve a similar goal, but also accomplish a better reconstruction on certain sequence classes than the Y chromosomal sequences derived from a long-read whole genome shotgun assembly with the current state of the art"*)

- P7,152: I assume this "read depth" approach refers to the Illumina-data-based CNV calling described in Methods. Is this increased copy number reflected in the assemblies? If that should not be the case, could the authors comment on the (potential) reasons. If the assemblies show multiple copies consistent with the CNV calls (and hence reveal the precise architecture), that would be a nice addition to the paper.

The read-depth approach refers to the CNV calling described in the methods. This is now explicitly stated in the manuscript. We have discussed the resolution of gene-families in the comparative annotation section, and in the supplementary Tables S10-S11. We included a statement in the manuscript regarding to challenges to fully reconstruct the genomic architecture of these copy number variations (*'While these expansions are to some degree represented in our assembly, the precise genomic architecture remains challenging to reconstruct. Due to the high degree of similarity between copies, several of them will be collapsed in the assembly specially in the AZFc region'*)

- The data access section still contains place holders, so I could look at the data.

We have filled in the placeholders; all data is now publicly available at the European Nucleotide Archive (ENA) under the study accession PRJEB28143. The assembly is deposited at the ENA under the accession ERZ678623.

- P7: "We manually validate all structural variants called in HG02982 in the 1000 Genomes Project in our assembly (see supplementary material)." In the supplement, I see three IGV screen shots. Can I conclude from that that 1KG made three SV calls? I think the authors should make more explicit what they mean by "validating" a call. In all three cases, Illumina data, 1000G calls, and calls made from the genome assembly do not seem to be in very good agreement. Working out the reasons for these differences would be important to develop a better understanding of the strengths/limitations of the proposed approach.

We have rephrased this sentence to now state 'We manually confirm the presence of all structural variants called in HG02982 in the 1000 Genomes Project in our data by checking the overlap of calls produced by orthogonal approaches'

The reviewer's assumption about the number of SV calls in the 1000 Genomes Project is correct. Generally, the discordance between the different the breakpoints is a matter of resolution. Read depth approaches, as used for the 1000 Genomes Project, are limited by their window size, which in term is limited by coverage. This underlines the importance of having complete assemblies, as the call of an SV – in theory -becomes as easy as looking at alignments.

- The fact that methylation status can be inferred due to using native DNA increases the appeal of this approach. However, the analysis of methylation levels is somewhat shallow and the reported correlation levels. In my view, some sort of control experiment would be needed to establish the reliability of the methylation calls.

We have now completely reworked the section regarding the methylation status. As the requested control experiment, we have produced whole genome bisulfite sequencing data for the same cell line and compared the concordance of the methylation status at CpG sites between the two call sets. We find the calls to be correlated with a pearson's r of 0.82 (see plot below). We now also asses the methylation status of genic regions within the different sequence classes of the Y chromosome, and find transcription start sites (TSS) of genes within the X-degenerate, the X-transposed and the pseudo-autosomal regions to exhibit low methylation levels, consistent with their broad expression patterns across the body Conversely, our results suggest high methylation levels in TSS of ampliconic genes, consistent with their specific expression in testis and silencing in the lymphoblastoid cell line used in this project.

- The paper is mostly technological and does not include much biological interpretation of the assembled sequence.

This manuscript is indeed intended as a methodological description of our approach, and thus technical.

- The ONT technology comes with the option of "read until", i.e. of using the sequencing device itself to selectively eject unwanted molecules from the nanopores, thereby enriching for particular sequences (e.g. doi:10.1038/nmeth.3930). I wonder whether the authors can comment on whether they considered to use this feature instead of the flow sorting and share their view on the feasibility.

Read until is a promising new method that takes advantage of ONT real-time processing to perform selective target enrichment of sequences of interest. Nevertheless, it is not applicable to our purposes, for the following reasons: First: The current implementation is applicable to maximum genome size of 5Mb only, a value several orders of magnitude smaller than the human genome. Additionally, the maximum captured target space in the aforementioned manuscript is 45 Kb and is unlikely that a target space as large as the Y chromosome can efficiently be queried in real-time at this moment, due to limitations in pore speed and computational resources.

Second, read until relies on a known target space for enrichment. While this might not be an issue for human Y chromosomes, which have a reference sequence available, our approach works for any chromosome that is sufficiently distinguishable in a flow-karyogram, without the need of prior knowledge of the underlying sequence.

- In Supp. Fig. 6, I do not see a coverage curve. Presumably something went wrong here with embedding the figure in the document? For this reason, I could not assess this drop in coverage at the PAR-1 boundary mentioned in the text. Could the missing genes (P6,L140) be due to the reduced coverage? In general, the coverage levels are at the low end of what one would want for genome assembly (and generation of a consensus), especially from noisy ONT data. I suspect that the assembly quality would go up quite a bit at an increased coverage.

We apologize for this inconvenience. There has been a rendering issue with this plot on some platforms, in which the data points were removed from the plot in the manuscript. We have replaced it with a correctly rendered version.

We do indeed believe that the shortcomings in the PAR-1 region are due to insufficient coverage, as implied by our reasoning on P5,L114ff (original submission). We hypothesize that the PAR-1 should be properly resolved with increased coverage, as it is not a particularly complex region taken out of the context of the whole genome (i.e. when assembled with the X chromosome). Unfortunately, we are not able to sequence deeper as the recovery of flow sorted chromosomes is low, and our initial calculation for the necessary amount of sorted chromosomes were based on higher yields than we achieved in some runs.

The missing genes are artifacts of the low coverage, which is stated in the manuscript.

Indeed, the coverage levels are at the minimum requirement of a canu assembly. Unfortunately, the recovery from flow sorting and hence the sequencing yields

- Figure 2: Panel 2: it is hard to judge whether the breaks in the dot plot coincide with the region boundaries. I suggest to color the background of the dot plot according to the region. Panel C: Colors for Insertion and Repeat_contraction are hard to distinguish (at least in my printout). Please add axis labels.

We have included a background coloring of the dotplot as per the reviewer's suggestions, showing (potential) coincidences of breakpoints with sequence class regions boundaries. We have also decided to replace panel 2C and moved it to the supplementary. Below is a rendering with a different color palette which we hope will increase distinguishability for the reviewer.

- P24,L472ff. This is duplicate and appears again on Page 26.

We have removed the unintentional duplicate method description and only kept the WGS assembly description

- P29,L585: likelyhood -> likelihood

We have corrected this typo

REVIEWERS' COMMENTS:

Reviewer #1 (Remarks to the Author):

The reviewers have addressed the majority of my comments satisfactorily and in a lot of detail.

In general, the paper is interesting and represents an advance over other methods. However I am concerned at the following comment:

"We now explicitly state what algorithm was used for the basecalling used in the assembly (MinKNOW 1.7.10 using Albacore 1.1). This was the default basecaller available by the time of data production. While we agree that ideally all data would have been re-called using the latest version now available, the choice of basecalling algorithms is a moving target and all analysis are based on this assembly. "

A specific benefit of Nanopore sequencing is the ability to recall data using the latest methods. Furthermore the difference between version 1.1 and 2.0+ is not insignificant. Albacore 2.0 introduces basecalling from raw signal rather than segmented events. If the goal of the paper is to provide a "reference-grade" assembly then surely this should be the best available to the authors at the time of writing. I am unclear on the reluctance to do this as the observations clearly do not reflect the state of the art even within the manuscript. If you recall the data why not use the recalled data?

However, my original comment only requested the authors clarify these details and they have done so.

Reviewer #2 (Remarks to the Author):

I appreciate all of the work that the authors have put in to improving the manuscript - particularly with respect the methylation portion, which was the area I felt needed the most work. The Illumina WGBS addition helps tremendously and brings the work to a satisfactory level of analysis and evaluation. The additional methods were also appreciated. I also agree that the scatterplot does not look as compelling or intuitive for the QC figure and agree with the authors to leave it as is, but I appreciate the effort to see if it helped.

In its current form, I believe it is suitable for publication and expect it will be of great interest to many in the genomics community.

Reviewer #3 (Remarks to the Author):

With the revisions provided, I am satisfied that the manuscript is suitable for publication.

Reviewer #4 (Remarks to the Author):

The authors have addressed many of the concerns of the reviewers. In particular the inclusion of additional WGBS data has strengthened the manuscript. In general, I think the manuscript is now publishable in Nature Communications, given that the authors address some few remaining points (detailed below). In particular, they should avoid overstating the benefits over assemblies done from

WGS data (which, in my view, are mostly in substantial cost savings). The study remains a technological piece and does not go into the biological interpretation of the assembled sequences. This is fine from my point of view since this technique might indeed enable studies on Y chromosome biology in the future.

Detailed responses to the authors' answers

INITIAL REVIEWER COMMENT: The authors cite Tomasziewicz et al. (2016) for an Gorilla Y chromosome assembly. In this context, it might be worth noting that the chimpanzee assembled as part of the recent Kronenberg et al. paper (Science, 2018) was male, so it would be interesting to compare to that assembly.

AUTHOR RESPONSE: We have included the comparison Tomasziewicz et al. (2016) solely due to the fact that it is based on a similar methodology, upon which we improve. However, we have performed a comparison to Y chromosomal sequences recovered from a whole genome shotgun assembly from NA24385 (Human, Ashkenazi son), which constitutes a similar case to the Chimp.

INITIAL REVIEWER COMMENT: While it makes sense to compare to existing assemblies, I would be cautious in claiming that the assemblies resulting from the strategy developed in this paper are in general superior to WGS assemblies. In my view, the main advantage lies in the cost-effectiveness for Y chromosome studies.

AUTHOR RESPONSE: We have rephrased our conclusion statement regarding sequence class completeness to state that the comparison to WGS only holds for this project and the current state of the art ("We show that we not only outperform previous efforts that sought to achieve a similar goal, but also accomplish a better reconstruction on certain sequence classes than the Y chromosomal sequences derived from a long-read whole genome shotgun assembly with the current state of the art")

REVIEWER RESPONSE: This sentence is vague and might therefore mislead readers. Please say specifically what you refer to by "...to achieve a similar goal", by "certain sequence classes", and by "with the current state of the art". Still, the authors suggest a superiority to WGS assemblies in general also in other places in the manuscript (e.g. P7, L141, where they refer to "WGS assemblies", in plural, rather than to this one assembly they compared to here). In my view, it is not necessary for this manuscript make this general statement. But if the authors absolutely want to retain this claim in the manuscript, they need to back this up by a much more thorough comparison to existing assemblies, including the Chimp from Kronenberg et al, the data from Jain et al., and potentially also the cliveome to the extent permitted by copyright restrictions. Also the Human Genome Structural Variation Consortium (HGSVC) has made PacBio data for three trios (e.g. including one father genome) publicly available (see Chaisson et al., 2017, bioRxiv).

INITIAL REVIEWER COMMENT: "We manually validate all structural variants called in HG02982 in the 1000 Genomes Project in our assembly (see supplementary material)." In the supplement, I see three IGV screen shots. Can I conclude from that that 1KG made three SV calls? I think the authors should make more explicit what they mean by "validating" a call. In all three cases, Illumina data, 1000G calls, and calls made from the genome assembly do not seem to be in very good agreement. Working out the reasons for these differences would be important to develop a better understanding of the strengths/limitations of the proposed approach.

AUTHOR RESPONSE: We have rephrased this sentence to now state 'We manually confirm the

presence of all structural variants called in HG02982 in the 1000 Genomes Project in our data by checking the overlap of calls produced by orthogonal approaches". The reviewer's assumption about the number of SV calls in the 1000 Genomes Project is correct. Generally, the discordance between the different the breakpoints is a matter of resolution. Read depth approaches, as used for the 1000 Genomes Project, are limited by their window size, which in term is limited by coverage. This underlines the importance of having complete assemblies, as the call of an SV – in theory -becomes as easy as looking at alignments.

REVIEWER RESPONSE: Please include the observation that 1000G has only made three calls in the manuscript.

INITIAL REVIEWER COMMENT: In Supp. Fig. 6, I do not see a coverage curve. Presumably something went wrong here with embedding the figure in the document? For this reason, I could not assess this drop in coverage at the PAR-1 boundary mentioned in the text. Could the missing genes (P6,L140) be due to the reduced coverage? In general, the coverage levels are at the low end of what one would want for genome assembly (and generation of a consensus), especially from noisy ONT data. I suspect that the assembly quality would go up quite a bit at an increased coverage.

AUTHOR RESPONSE: We apologize for this inconvenience. There has been a rendering issue with this plot on some platforms, in which the data points were removed from the plot in the manuscript. We have replaced it with a correctly rendered version. We do indeed believe that the shortcomings in the PAR-1 region are due to insufficient coverage, as implied by our reasoning on P5,L114ff (original submission). We hypothesize that the PAR-1 should be properly resolved with increased coverage, as it is not a particularly complex region taken out of the context of the whole genome (i.e. when assembled with the X chromosome). Unfortunately, we are not able to sequence deeper as the recovery of flow sorted chromosomes is low, and our initial calculation for the necessary amount of sorted chromosomes were based on higher yields than we achieved in some runs. The missing genes are artifacts of the low coverage, which is stated in the manuscript. Indeed, the coverage levels are at the minimum requirement of a canu assembly. Unfortunately, the recovery from flow sorting and hence the sequencing yields.

REVIEWER RESPONSE: The way this is presented is somewhat confusing. On the one hand, the authors say "Due to the sharp coincidence with the PAR-1 boundary, we do not suspect that the reduced coverage is a technical artifact of library preparation". On the other hand, they say "Mapping the raw data onto GRCh38 shows that this is an artifact, presumably due to insufficient coverage in this region.". These two sentences do not fit very well together. Wouldn't it be honest to say that the coverage was lower in the PAR region due to unknown reasons?

Minor comments

- If that is allowed by Nature Communications' policies, the authors should specifically state to which part of the Supplementary Material they refer to in each instance.

- P6, L107: maybe say "amounting to 21.5Mb of total sequence" ("1.46Mb amounting to 21.5Mb" sounds confusing)

- P9, L181: please state the fraction of CpGs that meet this criterion

- P11, L219: remove "with"

- P12, L224: ":", "  ":", "

Reviewer #4 (Remarks to the Author):

The authors have addressed many of the concerns of the reviewers. In particular the inclusion of additional WGBS data has strengthened the manuscript. In general, I think the manuscript is now publishable in Nature Communications, given that the authors address some few remaining points (detailed below). In particular, they should avoid overstating the benefits over assemblies done from WGS data (which, in my view, are mostly in substantial cost savings). The study remains a technological piece and does not go into the biological interpretation of the assembled sequences. This is fine from my point of view since this technique might indeed enable studies on Y chromosome biology in the future.

Detailed responses to the authors' answers

INITIAL REVIEWER COMMENT: The authors cite Tomaszekiewicz et al. (2016) for an Gorilla Y chromosome assembly. In this context, it might be worth noting that the chimpanzee assembled as part of the recent Kronenberg et al. paper (Science, 2018) was male, so it would be interesting to compare to that assembly.

AUTHOR RESPONSE: We have included the comparison Tomaszekiewicz et al. (2016) solely due to the fact that it is based on a similar methodology, upon which we improve. However, we have performed a comparison to Y chromosomal sequences recovered from a whole genome shotgun assembly from NA24385 (Human, Ashkenazi son), which constitutes a similar case to the Chimp.

INITIAL REVIEWER COMMENT: While it makes sense to compare to existing assemblies, I would be cautious in claiming that the assemblies resulting from the strategy developed in this paper are in general superior to WGS assemblies. In my view, the main advantage lies in the cost-effectiveness for Y chromosome studies.

AUTHOR RESPONSE: We have rephrased our conclusion statement regarding sequence class completeness to state that the comparison to WGS only holds for holds for this project and the current state of the art ("We show that we not only outperform previous efforts that sought to achieve a similar goal, but also accomplish a better reconstruction on certain sequence classes than the Y chromosomal sequences derived from a long-read whole genome shotgun assembly with the current state of the art")

REVIEWER RESPONSE: This sentence is vague and might therefore mislead readers. Please say specifically what you refer to by "...to achieve a similar goal", by "certain sequence classes", and by "with the current state of the art". Still, the authors suggest a superiority to WGS assemblies in general also in other places in the manuscript (e.g. P7, L141, where they refer to "WGS assemblies", in plural, rather than to this one assembly they compared to here). In my view, it is not necessary for this manuscript make this general statement. But if the authors absolutely want to retain this claim in the manuscript, they need to back this up by a much more thorough comparison to existing assemblies, including the Chimp from Kronenberg et al, the data from Jain et al., and potentially also the cliveome to the extent permitted by copyright restrictions. Also the Human Genome Structural Variation Consortium (HGSVC) has made PacBio data for three trios (e.g. including one father genome) publicly available (see Chaisson et al., 2017, bioRxiv).

We have rephrased the sentence to the following, to be more specific with what we refer to regarding the parts the reviewer felt are vague. We omit the use of "state of the art", as the dataset and methods are clearly described elsewhere in the manuscript:

"We show that we not only outperform previous efforts that sought to achieve a similar goal of reconstructing Y chromosomes but also accomplish a better reconstruction on all sequence classes than the Y chromosomal sequences derived from a long-read whole genome shotgun assembly."

We have also removed the sentence on P7 L141 claiming superiority over WGS assemblies.

INITIAL REVIEWER COMMENT: "We manually validate all structural variants called in HG02982 in the 1000 Genomes Project in our assembly (see supplementary material)." In the supplement, I see three IGV screen shots. Can I conclude from that that 1KG made three SV calls? I think the authors should make more explicit what they mean by "validating" a call. In all three cases, Illumina data, 1000G calls, and calls made from the genome assembly do not seem to be in very good agreement. Working out the reasons for these differences would be important to develop a better understanding of the strengths/limitations of the proposed approach.

AUTHOR RESPONSE: We have rephrased this sentence to now state 'We manually confirm the presence of all

structural variants called in HG02982 in the 1000 Genomes Project in our data by checking the overlap of calls produced by orthogonal approaches". The reviewer's assumption about the number of SV calls in the 1000 Genomes Project is correct. Generally, the discordance between the different the breakpoints is a matter of resolution. Read depth approaches, as used for the 1000 Genomes Project, are limited by their window size, which in term is limited by coverage. This underlines the importance of having complete assemblies, as the call of an SV – in theory - becomes as easy as looking at alignments.

REVIEWER RESPONSE: Please include the observation that 1000G has only made three calls in the manuscript.

We now state that the 1000G project has made 3 calls for this individual.

INITIAL REVIEWER COMMENT: In Supp. Fig. 6, I do not see a coverage curve. Presumably something went wrong here with embedding the figure in the document? For this reason, I could not assess this drop in coverage at the PAR-1 boundary mentioned in the text. Could the missing genes (P6,L140) be due to the reduced coverage? In general, the coverage levels are at the low end of what one would want for genome assembly (and generation of a consensus), especially from noisy ONT data. I suspect that the assembly quality would go up quite a bit at an increased coverage.

AUTHOR RESPONSE: We apologize for this inconvenience. There has been a rendering issue with this plot on some platforms, in which the data points were removed from the plot in the manuscript. We have replaced it with a correctly rendered version. We do indeed believe that the shortcomings in the PAR-1 region are due to insufficient coverage, as implied by our reasoning on P5,L114ff (original submission). We hypothesize that the PAR-1 should be properly resolved with increased coverage, as it is not a particularly complex region taken out of the context of the whole genome (i.e. when assembled with the X chromosome). Unfortunately, we are not able to sequence deeper as the recovery of flow sorted chromosomes is low, and our initial calculation for the necessary amount of sorted chromosomes were based on higher yields than we achieved in some runs. The missing genes are artifacts of the low coverage, which is stated in the manuscript. Indeed, the coverage levels are at the minimum requirement of a canu assembly. Unfortunately, the recovery from flow sorting and hence the sequencing yields.

REVIEWER RESPONSE: The way this is presented is somewhat confusing. On the one hand, the authors say "Due to the sharp coincidence with the PAR-1 boundary, we do not suspect that the reduced coverage is a technical artifact of library preparation". On the other hand, they say "Mapping the raw data onto GRCh38 shows that this is an artifact, presumably due to insufficient coverage in this region.". These two sentences do not fit very well together. Wouldn't it be honest to say that the coverage was lower in the PAR region due to unknown reasons?

We have rephrased the sentence from "Due to the sharp coincidence with the PAR-1 boundary, we do not suspect that the reduced coverage is a technical artifact of library preparation" to "We observe to drop off in coverage to coincide sharply with the PAR-1 boundary", and hope this is less confusing. Actually, we hypothesize that what we observe are unfinished recombination events, as we are sorting mitotic cells and recombination on the Y chromosome occurs only surrounding the PAR boundary. In the case of an unfinished recombination event, the PAR-1 would not co-sort with the Y chromosome in the flow cytometer, which in turn would lead to a decrease in coverage in that region.

Minor comments

- If that is allowed by Nature Communications' policies, the authors should specifically state to which part of the Supplementary Material they refer to in each instance.

We have included specific references to the corresponding part of the supplementary material

- P6, L107: maybe say "amounting to 21.5Mb of total sequence" ("1.46Mb amounting to 21.5Mb" sounds confusing)

We have rephrased the sentence according to the reviewer's suggestions.

- P9, L181: please state the fraction of CpGs that meet this criterion

We now state the number of CpG's that have minimum of 10X coverage in either dataset (n=4654).

- P11, L219: remove "with"

We have removed the typo

- P12, L224: ".,"  ","

We have removed the “.”